# ROOT CAUSE ANALYSIS OF FAILURE WITH OBSERVATIONAL CAUSAL DISCOVERY

## ABSTRACT

Finding the root cause of failures is a prominent problem in many complex networks. Causal inference provides us with tools to address this problem algorithmically to automate this process and solve it efficiently. The existing methods either use a known causal structure to identify root cause via backtracking the changes, or ignore the causal structure but rely on invariance tests to identify the changing causal mechanisms after the failure. We first establish a connection between root cause analysis and the *Interactive Graph Search (IGS)* problem. This mapping highlights the importance of causal knowledge: we demonstrate that any algorithm relying solely on marginal invariance tests to identify root causes must perform at least $\Omega(\log_2(n) + d\log_{1+d} n)$ many tests, where $n$ represents the number of components and $d$ denotes the maximum out-degree of the graph. We then present an optimal algorithm that achieves this bound by reducing the root cause identification problem as an instance of IGS. Moreover, we show that even if the causal graph is partially known in the form of a Markov equivalence class, we can identify the root-cause with linear number of invariance tests. Our experiments on a production-level application demonstrate that, even in the absence of complete causal information, our approach accurately identifies the root cause of failures.

## 1 INTRODUCTION

Root Cause Analysis (RCA) which aims to understand the root cause of failures is crucial for ensuring the reliability and stability of production systems in diverse domains, including but not limited to medicine (Kellogg et al., 2016; Latino, 2015), telecommunications (Schaaf et al., 2015), and IT operations (Whitney & Daniels, 2013; Drasar & Jirsik, 2019). In cloud applications, particularly those using microservice architectures, the challenges of RCA are even more pronounced. The large number of microservices complicates pinpointing the primary cause of failures (Emmons et al., 2022), and the interdependent nature of these services means that a failure in one can cascade, disrupting the entire network. These factors make timely and accurate diagnosis of failures particularly difficult. According to Wang et al. (2018),identifying the root cause of issues in platforms like IBM's Bluemix can take an average of three hours without automated tools. Therefore, rapid fault detection is essential for minimizing the downtime and mitigating the impact on system performance. Delays in diagnosing issues can lead to substantial financial losses and customer dissatisfaction, especially as service-level agreements often prioritize system availability as a key performance indicator.

Recent RCA research has focused on developing methods that detect the root cause of failures, often through a two-phase process: first, constructing a graph structure, then ranking the nodes within that graph. Some approaches rely on expert knowledge to build the graph (Ma et al., 2020), while others derive it from service call graphs (Chakraborty et al., 2023), or employ deep neural networks for graph learning (Lin et al., 2024). The goal is to model relationships and dependencies between services using causal discovery techniques to construct a causal graph (Wang et al., 2018; Qiu et al., 2020; Gan et al., 2021; Ikram et al., 2022; Xin et al., 2023). For instance, MicroCause (Meng et al., 2020) employs the PC algorithm to learn a causal graph from service metrics; however, the resulting graph is often an equivalence class with undirected edges, prompting researchers to arbitrarily convert it into a DAG. RUN (Lin et al., 2024), for example, removes the edge between two nodes with the lowest correlation, but this method does not guarantee the representation of the true underlying graph. In the second phase, existing algorithms rank nodes using graph centrality measures such as random walk (Wang et al., 2018; Ma et al., 2020), PageRank (Wu et al., 2021; Xin et al.,

2023; Lin et al., 2024), BFS (Lin et al., 2018), and DFS (Chen et al., 2014). However, many rely on arbitrary objective functions that may not accurately reflect the failure propagation chain. For example, Groot (Wang et al., 2021) assumes that sink nodes are more likely to be root causes and assigns them a different score than others.

In causal discovery-based approaches, Ikram et al. (2022) recently observed that a fault alters the generative mechanism of the failing node. This observation frames the fault as an intervention on the node, classifying the data generated during the failure period as a post-interventional dataset. Building on this idea, the authors introduced RCD (Root Cause Discovery), which leverages established techniques to identify the interventional target *i.e.*, the root cause of the failure. RCD does not learn the causal graph but only uses conditional independence (CI) tests to find the interventional target. An example demonstrating the execution of RCD is provided in the Appendix E.

Despite the existing literature, we assert that current RCA methods overlook a crucial opportunity: they fail to utilize the system's normal operation time. While identifying the root cause of a failure is a time-sensitive task *once the failure occurs*, the period *before the failure* offers ample time for preparation. During normal operations, site engineers or RCA systems can proactively prepare for potential failures by learning cause-effect relationships through domain knowledge or causal discovery from observational data, a topic extensively explored in the literature (Spirtes & Glymour, 1991; Spirtes et al., 2000; Chickering, 2002; Peters et al., 2013; Zheng et al., 2018; Lam et al., 2022). In this context, observational data refers to metrics collected before a failure occurs, while post-interventional data pertains to metrics gathered after the failure.

**Our contribution.** In this paper, we introduce a novel algorithm, **Root Cause Analysis with Causal Graphs (RCG)**, which strategically utilizes a system's normal operational period to prepare for potential failures. We propose learning a causal graph from observational data collected during regular operations. This learned graph is then used proactively to determine which invariance tests should be conducted post-failure. We demonstrate how to integrate this causal knowledge into RCA without relying on arbitrary assumptions about the system's structure. We begin by exploring the simplest case, where the causal relationships are fully known—that is, when the causal graph is a DAG. Interestingly, we show that identifying the root cause in a causal DAG is equivalent to solving a well-established graph theory problem known as Interactive Graph Search (IGS) (Tao et al., 2019), with minor modifications. This reduction to IGS not only enables us to leverage its logarithmic computational complexity but also establishes a lower bound on the number of CI tests required.

Learning the full causal DAG of a system is often challenging in real-world scenarios. To address this, we investigate how to leverage a partial causal structure derived from data for RCA. Instead of arbitrarily converting a partial causal graph to a DAG, we propose a systematic approach to traverse the graph structure for root cause identification. Additionally, we note that existing causality-based methods, such as RCD (Ikram et al., 2022), typically rely on higher-order CI tests, which involve testing with large conditioning sets. This can significantly diminish statistical power, especially with finite sample sizes. Although RCD attempts to reduce this issue by partitioning nodes into smaller subsets, it does not guarantee a meaningful decrease in the number of required CI tests.

To mitigate these challenges, we try to minimize the use of higher-order CI tests by limiting the size of the conditioning sets (Spirtes, 2001; Rohekar et al., 2021). Our approach uses the $\mathcal{C}$-PC algorithm (Lee et al., 2024), which constrains conditioning set sizes to learn a partial causal graph, thereby reducing errors from limited statistical power in CI tests on finite samples. Furthermore, we propose an algorithm that leverages this partial causal graph to identify the root cause of failures. Consequently, we demonstrate that even with incomplete graph knowledge, it is possible to accurately pinpoint the root cause by using at most a linear number of marginal invariance tests.

1. For the case of having a single root cause, given a complete causal structure of a system, we first map the problem of RCA to IGS and then further provide an algorithm that identifies the root cause with $\mathcal{O}(\log_2(n) + d \log_{1+d} n)$ number of marginal invariance tests and show that any algorithm that solely relies on marginal invariance tests for RCA must perform $\Omega(\log_2(n) + d \log_{1+d} n)$ many tests.

2. In scenarios where only observational data is available, we consider the challenge of learning a partial causal structure from the system's data-generating process. We propose an algorithm that leverages this estimated graph structure and an information-theoretic ap-

proach to pinpoint multiple root causes of failures. We further demonstrate that, given an estimated causal structure, the proposed algorithm is theoretically sound for RCA.

3. We validate the performance of our proposed algorithm by showing its comparable accuracy relative to state-of-the-art methods, such as RCD (Ikram et al., 2022), RUN (Lin et al., 2024), and BARO (Pham et al., 2024), through experiments on simulated datasets, including the Sock-shop dataset (Holbach., 2022), and a production-level application.

## 2 BACKGROUND

In this section, we give the most relevant definitions. For more details of the graph notations and terminology, please refer to Appendix A. We also discuss related work in Appendix B.

**Definition 2.1** (Causal graphs). A *causal graph* is used to encapsulate the causal relationships among variables in the form of a directed acyclic graph (DAG), where each node represents a variable $X$ and the directed edge $X \to Y$ indicates that $X$ causes $Y$. A variable is said to cause another variable if a change in the former induces a change in the probability distribution of the latter.

**Structural Causal Models (SCMs) and Causal Bayesian Networks (CBNs).** SCMs are used to model causality among a set of random variables. Each variable $X$ is a function of some endogenous variables as its parents, denoted by $Pa(X)$, and an exogenous noise term, denoted as $E_X$ e.g. $X = f_X(Pa(X), E_X)$. An SCM induces a causal graph by assigning a set of endogenous variables as the parents of $X$ for all variables $X$. CBNs are used to define a causal model that specifies the observational and interventional distributions via the truncated factorization formula without the functional descriptions like SCMs in a causal graph.

**Definition 2.2** (d-separation). In a causal graph $D$, a path $p$ between $X$ and $Y$ is *d-connecting (active)* relative to a set of vertices $\mathbf{Z}(X, Y \notin \mathbf{Z})$ if $(i)$ every non-collider on $p$ is not in $\mathbf{Z}$ and $(ii)$ every collider on $p$ is an ancestor of some $Z \in \mathbf{Z}$. Otherwise, we say $\mathbf{Z}$ *blocks* $p$. If $\mathbf{Z}$ blocks all paths between $X$ and $Y$, we say $X$ and $Y$ are *d-separated* relative to $\mathbf{Z}$, denoted as $(X \perp\!\!\!\perp Y | \mathbf{Z})_D$.

**Intervention and F-NODE.** An intervention on a variable is the process of changing the generative mechanism of that variable. Randomized controlled trials (RCTs) and A/B tests are the most common notion of interventions. Pearl uses do-operator $do(X = x)$ to capture this type of intervention. For instance, when $do(X = x)$ forces a variable $X$ to take on certain values, it is known as the *hard* interventions (Pearl, 2009). Its effect in a causal graph is to remove the edges incoming to the intervened nodes. It is different than another type of intervention known as the *soft interventions*, which do not completely alter the causal mechanisms and retain the original causal graph by only replacing $f_X(Pa(X), E_X)$ with $f'_X(Pa(X), E_X)$ where $f' \neq f$. A variable F-NODE has been extensively used to represent the effect of an intervention on a system (Pearl, 1995; Yang et al., 2018; Mooij et al., 2020). Throughout this work, we denote a ground truth DAG $D$ being augmented by F-NODE as an intervention to the root cause as $D_{aug}$. We assume the extended faithfulness assumption as in Jaber et al. (2020). Please refer to Appendix A.15 and A.16 for more details.

## 3 PROBLEM FORMULATION

A system has $n$ components $\mathcal{M} = \{m_1, \ldots, m_n\}$. Within a given time interval, the monitoring tool collects at least $d$ metrics from each of the components, i.e. $\mathcal{T}(i, t) = \{r_{i,1,t}, \ldots, r_{i,d,t}\}$, where $d \geq 1; \forall i \in \{1, \ldots, n\}$, $\mathcal{T}(i, t)$ is a set of $d$ metrics of component $i$ at time instance $t$. Considering the entirety of the data, we have two time series datasets defined as $\mathcal{D} = \{\mathcal{T}(1, 1), \ldots, \mathcal{T}(n, t-1)\}$ and $\mathcal{D}^\star = \{\mathcal{T}(1, t), \ldots, \mathcal{T}(n, \gamma)\}$, where $t$ represents the time when the failure was first registered and $\gamma$ is the time when the issue was fixed. We consider the setting where one can learn some cause-effect relations in the form of a $\mathcal{C}$-essential graph[1] $\varepsilon_{\mathcal{C}}(D) = (\mathbf{V}, \mathbf{E})$ at the time $s$ from $\mathcal{D}$, where $s < t$ and $\mathcal{C}$ is the set of conditioning sets used for all CI tests, $\mathbf{V}$ denotes the set of $d$ metrics as random variables and $\mathbf{E}$ is the set of edges where $X_i \to X_j$ represents metric $X_i$ causes metric $X_j$. We leverage this partial causal structure to pinpoint the root cause between timestamps $t$ and $\gamma$.

**Failure as Interventions.** An important observation of this problem is to model a failure as a soft intervention on the failing mode (Ikram et al., 2022). Here, the representation of F-NODE allows one

---

[1] Please see Appendix A for the definition of $\mathcal{C}$-essential graphs.

to identify the distribution invariances $P_N(X|Pa(X)) = P_A(X|Pa(X))$, where $P_N$ and $P_A$ are the distributions under normal mode of operation and anomalous operation respectively. By concatenating both of these datasets, one can sample from the distribution $P^\star$ of a set of observed variables $\mathbf{V}$ involving F-NODE, denoted as F, where $P^\star(\mathbf{V}|F = 0) = P_N(\mathbf{V})$ and $P^\star(\mathbf{V}|F = 1) = P_A(\mathbf{V})$. Under this formalism, the invariance $P_N(X|Pa(X)) = P_A(X|Pa(X))$ corresponds to conditional independence between $X$ and $F$ given $Pa(X)$. Since F-NODE cannot have any incoming edges, one can then employ a series of CI tests on the sampling distributions $\hat{P}^\star$ to determine which node is the root cause $R$ (the child of F-NODE) by observing $(R \not\perp\!\!\!\perp F|Pa(R))_{\hat{P}^\star}$.

Performing an exponentially large number of CI tests, as required by RCD, is far from ideal in post-failure scenarios. This is due to the fact that RCD operates without any prior causal knowledge. RCD focuses solely on identifying the adjacency of the F-NODE rather than learning the entire graph, as constructing the full causal structure can be time-consuming. However, it is important to note that RCA is time-sensitive *only* after the failure occurs. The time leading up to a failure provides ample opportunity to prepare the system. Therefore, we propose leveraging this pre-failure window to learn the causal graph from observational data, which can then be used post-failure to effectively identify the root cause. In the following sections, we will first highlight the benefits of having complete causal knowledge of the underlying data-generating mechanism, followed by a more practical approach for cases where the causal graph is unknown.

## 4 RCA WITH A KNOWN GRAPH

In this section, we discuss the main limitation of RCD's approach as our approach also models failure as an intervention. Then, we introduce the use of graphical structures as a potential solution in the case of a single root cause. For details on RCD, see Appendix E and all proofs are provided in the Appendix C.

Firstly, RCD only learns the adjacencies between F-NODE and each observed variable as it operates. It conditions on every possible subset $\mathbf{S}$ of variables $\mathbf{V}$ for testing the conditional independence relation between each pair of variables i.e., $X, Y \in \mathbf{V}$ until it identifies a conditioning set that yields conditional independence, which excludes a potential node as the root cause under Assumption A.16. However, under Assumption A.15, having access to a causal graph $G$ allows us to conduct $n$ CI tests *e.g.*, $(F \perp\!\!\!\perp X|Pa_G(X))$ for each observed variable $X$ where $n$ is the number of observed variable. In other words, RCD performs at least as many CI tests as would be required in a naive approach using the causal graph. Secondly, RCD may condition on a set of variables that is much larger than the actual parent set, resulting in unreliable CI test results in practice. In contrast, since our graphical structure captures ancestral relationships between nodes and there is only a single root cause variable, we argue that the root cause can be identified with fewer than $n$ tests. To support this, we present key results that allow for a systematic exploration of the causal structure, significantly reducing the number of required CI tests.

For the case where there is only a single root cause, the following two lemmas indicate that certain CI relations can eliminate variables from being considered as root causes, under Assumption A.15 and A.16. The first lemma states that all ancestors of a variable $X$ can be excluded as the root cause if we observe that $F$ is conditionally independent of $X$ given some variables $\mathbf{Z}$. The second lemma asserts that all non-ancestors of $X$ can be excluded as the root cause if $F$ is conditionally dependent on $X$. Unlike RCD, which performs a series of CI tests and stops once a CI relation excludes a variable as the root cause, our approach systematically eliminates variables using these two key results—Lemma 4.1 and Lemma 4.2—without needing to test every variable. We provide an example to illustrate how these two lemmas may enable us to identify the root cause in fewer than $n$ tests given a causal graph in Appendix G.

**Lemma 4.1.** *Given a causal graph $D$, if $(F \perp\!\!\!\perp X)_P$ for some $X \in \mathbf{V}$, then $A \notin Ch_{D_{aug}}(F)$ for all $A \in An_D(X)$, where $P$ is any joint distribution between variables on $D_{aug}$.*

**Lemma 4.2.** *Given a causal graph $D$, if $(F \not\perp\!\!\!\perp X)_P$ for some $X \in \mathbf{V}$, then then $Q \notin Ch_{D_{aug}}(F)$ for all $Q \in NAn_D(X)$, where $P$ is any joint distribution between variables on $D_{aug}$.*

To illustrate the usefulness of these two key results, we show that there is a one-to-one correspondence between using the marginal invariance test for RCA with a known causal graph and the prob-

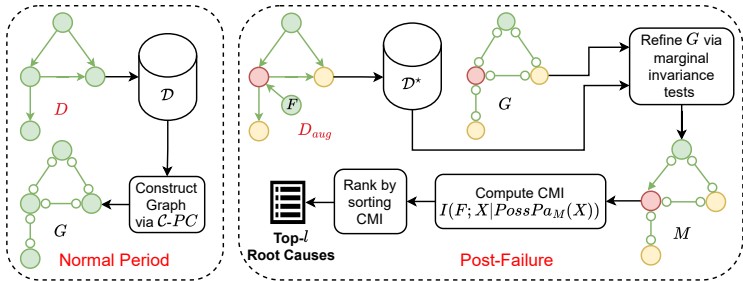

Figure 1: The RCG framework: The true graphs, $D$ and $D_{aug}$, are unknown to the algorithm. Red nodes represent the root cause, while orange nodes are impacted but not the root cause. During the normal period, RCG learns the partial causal graph $G$ from data using $\mathcal{C}$-PC. After a failure, it identifies the root cause by performing marginal invariance tests to further orient the edges and computing the Conditional Mutual Information (CMI) between the F and each node in the graph. Finally, RCG ranks the nodes by CMI scores, outputting an ordered list of potential root causes.

lem known as *Interactive Graph Search* (IGS) (Tao et al., 2019) that guarantees to identify the root cause with fewer than $n$ tests. For the sake of clarity, we give the problem formulation of IGS.

**Interactive Graph Search (IGS)**
INSTANCE: A DAG $D = (\mathbf{V}, \mathbf{E})$ that has a single root node, an adversary chooses arbitrarily a target node $R \in \mathbf{V}$. There is an oracle that returns a boolean answer to the given query: yes, if there is a directed path from $X$ to $R$ and no otherwise for any $X \in \mathbf{V}$.
QUESTION: What is the minimum number of queries to ask in order to identify $R$ in $D$?

**Lemma 4.3.** *Consider a DAG $D = (\mathbf{V}, \mathbf{E})$ with a single sink node and $D'$ be a DAG by reversing every edge direction in $\mathbf{E}$, let $Q(X)$ be a query to the oracle on whether some $X \in \mathbf{V}$ has a directed path to an unknown target node $R \in \mathbf{V}$.*

$$Q(X) = yes \Leftrightarrow (F \not\perp\!\!\!\perp X)_P \tag{1}$$

*. Therefore, if $Q(X) = yes$, then $X \in An_{D'}(R)$. If $Q(X) = no$, then $X \in NA_{D'}(R)$.*

The significance of Lemma 4.3 is that a solution to IGS is now a solution to RCA using the marginal invariance tests given a known causal graph. For DAGs that do not have a single sink node, we can simply add a dummy node as a child of all the sink nodes. Hence, the following theorem is an immediate consequence of Theorem 1 (see Appendix C.1) proven by Shangqi et al. (2023)).

**Theorem 4.4.** *Given a causal graph $D$ with a single sink node, any algorithm the only uses marginal invariance tests must perform $\Omega(\log_2 n + d\log_{1+d} n)$ many tests to find the single root cause in the worst case, where $d$ is the maximum in-degree of $D$ and $n$ is the number of nodes. There exists an algorithm that finds the root cause with $\mathcal{O}(\log_2 n + d\log_{1+d} n)$ marginal invariance tests.*

Shangqi et al. (2023) provide an optimal algorithm that bounds the number of queries in $\mathcal{O}(\log_2 n + d\log_{1+d} n)$ in the worst case for IGS. Due to Lemma 4.3, this algorithm can be modified for RCA with a single root cause using marginal invariance tests. Hence, we showed that we need fewer than $n$ tests and that marginal invariance tests alone are sufficient for identifying the root cause given a causal graph. We provide the pseudocode of modified IGS through Algorithm 7 in the Appendix.

## 5 RCA WITH AN UNKNOWN GRAPH

Having established that the graphical structure helps reduce the number of CI tests, we now turn our attention to the challenge of performing RCA with partial graphical structure in the case of multiple root causes. We provide the workflow of the proposed solution in Figure 1. We leave all the proofs in the Appendix C.

One common approach to learning the causal structure is to incorporate expert knowledge (Chakraborty et al., 2023; Gong et al., 2024; Lin et al., 2024; Xin et al., 2023). However,

it may not always be feasible to obtain expert knowledge. A data-driven approach to causal structure learning then becomes a more viable solution. However, learning a causal structure can be extremely time-consuming (Chickering et al., 2004). For constraint-based methods, they often involve conditioning on large sets of nodes to identify possible separating sets for each node (Spirtes et al., 2000). This time-consuming aspect of causal discovery is particularly undesirable in our context, where time is critical following a failure, and the goal is to quickly pinpoint the root cause.

A key point is that learning causal structures does not require interventional data (Spirtes et al., 2000; Chickering, 2002; Shimizu et al., 2006; Zheng et al., 2018). We can leverage the vast amounts of data generated during the system's normal operation to construct the causal graph, rather than waiting for a failure. This graph can then be used to efficiently identify the root cause when a failure occurs, enabling a faster, more effective response.

**Ranking Root Causes.** A key requirement for RCA tools is the output format. While failures typically have few root causes, much of the literature focuses on ranking all nodes and reporting the top-$l$. This poses a challenge for approaches that rely on CI tests, which often identify only a single or a few root cause nodes. RCD addresses this by gradually increasing the significance level, $\alpha$, in its CI tests and rerunning the algorithm until at least $l$ nodes are identified. However, this does not guarantee a meaningful ranking; the resulting nodes may appear in an arbitrary order, and multiple reruns increase runtime. To address this, RCG (Algorithm 1) leverages a critical insight that the ranking in RCA aligns with an information-theoretic approach shown by the following proposition.

**Proposition 5.1.** *Given any DAG $D$, under Assumptions A.15 and A.16,*

$$I(F; R|Pa_D(R)) > 0 \tag{2}$$
$$I(F; \bar{R}|Pa_D(\bar{R})) = 0 \tag{3}$$

*, where $R$ is the actual root cause and $\bar{R}$ denotes a non-root cause variable.*

The intuition behind proposition 5.1 is that any non-root-cause variable $\bar{R}$ must be d-separated from $F$ given its parents $Pa_{\bar{R}}$, while only the true root cause $R$ is d-connecting with $F$ given its parents $Pa(R)$. Under the faithfulness assumption, $F$ must be conditionally dependent with $R$ given $Pa(R)$, and by Causal Markov condition, $F$ must be conditionally independent with $\bar{R}$ given $Pa(\bar{R})$. These conditional independencies can be measured using CMI. Thus, RCA with unknown graph can be broken down into two steps: finding the parents of each variable and estimating the CMI given its parents. Ranking the potential root causes is then done by sorting the CMI values in descending order. This non-parametric method is robust, capturing both linear and nonlinear dependencies, and works across various types of distributions, whether discrete, continuous, or mixture.

Learning the partial causal graph from data requires a series of high-order CI tests (Spirtes et al., 2000). However, the statistical power of these tests diminishes significantly as the size of the conditioning set increases (Shah & Peters, 2020; Kocaoglu, 2023). To address this issue, we propose using a more robust approach through the generalized $\mathcal{C}$-PC algorithm (Lee et al., 2024), which obtains a $\mathcal{C}$-essential graph. This graph represents the Markov equivalence class of DAGs based on a restrictive set $\mathcal{C}$ of conditioning sets. The set $\mathcal{C}$ allows us to specify which conditioning set to use, enabling reliance on CI tests with smaller conditioning sets and avoiding high-dimensional variables. For details about the $\mathcal{C}$-essential graph and its interpretation, see Appendix A and F. We also discuss the challenges of using CI tests exclusively for RCA with a $\mathcal{C}$-essential graph in Appendix I. Our key contribution is that only $n$ marginal invariance tests need to be conducted during failure to obtain a superset of the parent set for each non-root-cause variable $\bar{R}$ that d-separates $\bar{R}$ from $F$, where $n$ is the number of observed variables. While Lemma 5.2 ensures the correctness of Algorithm 2, Lemma 5.3 connects Algorithm 1 with Proposition 5.1 through the use of possible parent sets.

**Lemma 5.2.** *Given a distribution $P$ defined over a set of CIs based on a conditionally closed set $\mathcal{C}$, for any $X, Y \in \mathbf{V}$ and $\mathbf{Z} \in \mathcal{C}$, if $(X \perp\!\!\!\perp Y |\mathbf{Z})_P, (X \not\perp\!\!\!\perp W |\mathbf{Z})_P$, then no DAG faithful to $P$ contains the edge $W \rightarrow Y$.*

**Lemma 5.3.** *Let $M$ be the graph returned by Algorithm 2, $F$ is not adjacent to $X$ in $D_{aug}$ if and only if $F$ is d-separated with $X$ given $PossPa_M(X)$ in $D_{aug}$.*

Next, we briefly discuss the trade-off between computational efficiency and sample complexity in Algorithm 1. As noted by Corollary 5.4, a larger set $\mathcal{C}$ allows the $\mathcal{C}$-PC algorithm to conduct more CI tests, potentially including high-order tests. While this tends to result in a sparser graph, it

**Algorithm 1** Root Cause Analysis with Causal Graphs (RCG)

**input** Observational data $\mathcal{D}$, interventional data $\mathcal{D}^\star$, a $\mathcal{C}$-essential graph $G$, a required number of root causes $l$.
**output** top $l$ root causes
1: $D \leftarrow$ Concatenate $\mathcal{D}$ and $\mathcal{D}^\star$ with $F$
2: $G \leftarrow$ **MARGINAL-INVARIANCE**$(D, G)$
3: **for** $X \in \mathbf{V}$ **do**
4:     $I_X \leftarrow$ Estimate $I(F; X | PossPa_G(X))$
5: **end for**
6: $\mathbf{V}_s \leftarrow$ Sort $X \in \mathbf{V}$ by $I_X$ in descending order
7: **Return** the first $l$ root causes from $\mathbf{V}_s$.

**Algorithm 2** MARGINAL-INVARIANCE

**input** Observational and interventional data distribution $P$, a $\mathcal{C}$-essential graph $G = (\mathbf{V}, \mathbb{E})$, CI tester.
**output** $G$
1: **for** $X, Y \in \mathbf{V}$ **do**
2:     **if** $(F \perp\!\!\!\perp X)_P$ and $(F \not\perp\!\!\!\perp Y)_P$ **then**
3:         If $X \leftarrow Y$ is in $G$, remove $X \leftarrow Y$
4:         If $X - Y$ is in $G$, orient $X \rightarrow Y$
5:         If $Xo\!\!-\!\!oY$ is in $G$, orient $Xo\!\rightarrow Y$
6:         If $X \leftarrow\!oY$ is in $G$, orient $X \leftrightarrow Y$
7:     **end if**
8: **end for**
9: **Return** $G$

also increases the time needed to learn the causal graph during normal operations and requires more samples for reliable CI tests. The goal is to reduce the set of possible parents during normal operation by conducting more informative CI tests based on data reliability. Although our method can leverage advancements in consistent CMI estimators for high-dimensional datasets (Mukherjee et al., 2020; Li et al., 2023), a smaller set of possible parents will reduce the time needed to compute CMI during critical failure situations. We provide more discussion on this topic in the Appendix H.

**Corollary 5.4.** *Given two graphs $M_1, M_2$ returned by Algorithm 2 based on two different $\mathcal{C}$-essential graphs $\varepsilon_{\mathcal{C}_1}(D)$ and $\varepsilon_{\mathcal{C}_2}(D)$, if $\mathcal{C}_1 \subset \mathcal{C}_2$, then $|PossPa_{M_1}(X)| \geq |PossPa_{M_2}(X)|$.*

## 6 EXPERIMENTS

In this section, we evaluate RCG's accuracy by addressing two key questions: 1) *Does a causal graph help RCG identify the root cause?* 2) *How quickly can RCG find the root cause?* We then discuss our implementation setup and present the results. We provide additional results in Appendix J.

**Implementation.** To generate experimental data, we followed a streamlined approach (Ikram et al., 2022; Lin et al., 2024), using `pyagrum` (Ducamp et al., 2020) to create random causal graphs. We then generated samples for both observational and interventional settings by perturbing the data generation process of a randomly selected node. To ensure robustness, each experiment was repeated 100 times, with results reported as mean and standard error. In RCA literature, a key metric for evaluating effectiveness is accuracy at top-$l$, defined as the probability of identifying the root cause within the top $l$ ranked causes. Hence, we report top-$l$ accuracy along with the execution runtime.

For our experiments, we implemented the following baselines:

- **RUN** (Lin et al., 2024): It constructs a causal graph using neural Granger causal discovery with contrastive learning. It ranks the nodes by PageRank with a personalized vector according to the learned graph.

- **MI**: A simple approach that sorts each node based on its mutual information with F.

- **RCD** (Ikram et al., 2022): A recent method that uses CI tests to identify the root cause.

- **RCG**: A prototype of Algorithm 1, which uses $\mathcal{C}$-PC to learn a causal graph. We use a postfix to indicate how $\mathcal{C}$ is chosen, so RCG-$k$ means that the input $\mathcal{C}$-essential graph to Algorithm1 was learned using $\mathcal{C}$-PC with $\mathcal{C}$ containing all conditioning sets of size up to $k$.

To demonstrate the value of graphical structure, we first present an experiment where all baselines used the ground truth graph as input. The results with graphs learned from data are shown in Appendix J.2. We also compare three variants of RCG: RCG(IGS)[2], which takes a DAG as input and identifies the root cause per Lemma 4.3; RCG-2; and RCG(CPDAG), which uses the essential graph

---

[2]For IGS, we referenced the recent findings from the POMS paper (Shangqi et al., 2023), but the authors declined to share their code in a way that can be made public. Consequently, we implemented an older, simpler version from (Tao et al., 2019). For a runtime comparison, please see Theorem C.1 and C.2 in Appendix.

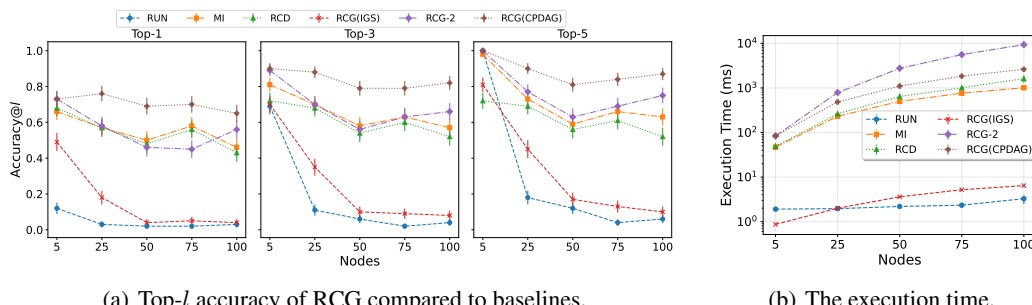

(a) Top-$l$ accuracy of RCG compared to baselines.    (b) The execution time.

Figure 2: The results demonstrate that RCG with RCG-2 consistently provides better accuracy compared to RCD. While MI struggles due to its inability to condition on the parents of each node, whereas RCD is capable of conditioning on other nodes but lacks information about the causal structure. In contrast, RCG overcomes these challenges by learning a causal graph and using CMI to rank the nodes effectively.

generated by the PC algorithm (Spirtes et al., 2000). Furthermore, we used 10,000 samples for the normal period and only 100 samples for the post-failure dataset.

Figure 2 shows the top-$l$ accuracy and runtime of different approaches with $l = 1/3/5$. Notably, the accuracy of RCG (IGS) declines sharply, despite offering the lowest runtime among all CI-based methods. This drop occurs because IGS assumes every query is perfect, but in our context, running a CI test can yield incorrect results depending on sample availability. Consequently, IGS makes erroneous decisions, resulting in poor performance as the number of nodes increases. This highlights that while IGS presents strong theoretical results, it struggles with imperfect CI tests, where a single error can lead to cascading failures. Similarly, RUN performs poorly due to its PageRank personalization algorithm, which incorporates arbitrary constraints not applicable to our experimental setup, such as assuming that leaf nodes are more likely to be the root cause. As a result, even with the ground truth DAG, RUN fails to identify the root cause.

Comparing RCD and RCG, we find that RCG-2 generally achieves better accuracy. With 100 nodes, RCG-2 identifies the root cause in the top-1 position with an accuracy of 56%, surpassing RCD's 43%. This can improve to 65% if an accurate essential graph is learned. The superior performance of RCG-2 stems from its sample-efficient graph learning using $\mathcal{C}$-PC. However, RCG-2 also exhibits the highest runtime because, with $k = 2$, it can only run CI tests where the size of the separating set is 2. As discussed in Section 5, controlling the size of the separating sets affects graph sparsity; smaller $k$ values lead to denser graphs. At $k = 2$, we observe a substantial number of spurious edges that could be removed by conditioning on a larger separating set. This increase the set of potential parents for each node, subsequently raising the runtime for CMI calculations.

Nonetheless, the runtime can be reduced if an accurate sparse graph is learned during normal operations. This is evident from the runtime of RCG (CPDAG), where the input essential graph is learned from $k = n - 2$ (with $n$ as the number of nodes). Thus, RCG offers a trade-off between the number of observational samples and the runtime for identifying the root cause post-failure. More observational samples result in a sparser graph, which increases runtime before the failure but ultimately reduces runtime *after* the failure.

## 7  CASE STUDY

**Sock-shop.** This section demonstrates the effectiveness of RCG using the Sock-shop application, a microservice-based replica of a web app for selling socks. The system consists of 13 microservices, with 5 being the most critical and user-facing. Although Sock-shop is microservice-based, our method remains system-agnostic. We used the dataset from Ikram et al. (2022), which includes two failure types: CPU hog and memory leak. The dataset contains 50 instances, each running for 5 minutes in both normal and failure conditions. Each experiment was repeated 50 times, and we report the mean top-$l$ accuracy.

| | | MI | cRCA | RUN | BARO | RCD | RCG-0 | RCG-C | RCG-1 | RCG-Expert |
|---|---|---|---|---|---|---|---|---|---|---|
| | Carts | 0.79 | 0.80 | 0.00 | 1.00 | 0.56 | 0.54 | 0.67 | 0.80 | 0.30 |
| | Catalogue | 0.11 | 0.40 | 0.00 | 1.00 | 0.18 | 0.97 | 0.43 | 0.10 | 0.81 |
| top-1 | Orders | 0.36 | 0.40 | 0.00 | 1.00 | 0.68 | 0.72 | 0.82 | 0.40 | 0.96 |
| | Payment | 0.27 | 0.40 | 0.00 | 1.00 | 0.65 | 0.78 | 0.84 | 0.29 | 0.93 |
| | User | 1.00 | 1.00 | 0.00 | 1.00 | 1.00 | 0.72 | 0.90 | 1.00 | 0.87 |
| | Avg. | 0.51 | 0.60 | 0.00 | 1.00 | 0.61 | 0.75 | 0.73 | 0.52 | 0.77 |
| | Carts | 1.00 | 0.80 | 0.42 | 1.00 | 0.87 | 0.55 | 0.74 | 1.00 | 1.00 |
| | Catalogue | 0.92 | 0.60 | 0.39 | 1.00 | 0.47 | 1.00 | 0.73 | 0.57 | 1.00 |
| top-3 | Orders | 1.00 | 0.40 | 0.07 | 1.00 | 0.92 | 0.73 | 0.85 | 1.00 | 1.00 |
| | Payment | 1.00 | 0.40 | 0.14 | 1.00 | 0.88 | 0.78 | 0.88 | 1.00 | 1.00 |
| | User | 1.00 | 1.00 | 0.06 | 1.00 | 1.00 | 0.78 | 0.94 | 1.00 | 1.00 |
| | Avg. | 0.98 | 0.64 | 0.22 | 1.00 | 0.77 | 0.75 | 0.83 | 0.91 | 1.00 |
| | Carts | 1.00 | 1.00 | 0.62 | 1.00 | 0.82 | 0.55 | 0.75 | 1.00 | 1.00 |
| | Catalogue | 0.93 | 0.60 | 0.58 | 1.00 | 0.51 | 1.00 | 0.89 | 0.82 | 1.00 |
| top-5 | Orders | 1.00 | 0.60 | 0.15 | 1.00 | 0.87 | 0.74 | 0.86 | 1.00 | 1.00 |
| | Payment | 1.00 | 0.40 | 0.20 | 1.00 | 0.86 | 0.78 | 0.88 | 1.00 | 1.00 |
| | User | 1.00 | 1.00 | 0.11 | 1.00 | 1.00 | 0.82 | 0.97 | 1.00 | 1.00 |
| | Avg. | 0.99 | 0.72 | 0.33 | 1.00 | 0.81 | 0.78 | 0.87 | 0.96 | 1.00 |

Table 1: The table shows the top-$l$ accuracy of different baselines on the data collected from sock-shop application after injecting CPU hog to a given microservice.

| | | MI | cRCA | RUN | BARO | RCD | RCG-0 | RCG-C | RCG-1 | RCG-Expert |
|---|---|---|---|---|---|---|---|---|---|---|
| | Carts | 0.87 | 0.20 | 0.02 | 1.00 | 0.58 | 1.00 | 1.00 | 0.87 | 0.36 |
| | Catalogue | 0.10 | 0.20 | 0.00 | 1.00 | 0.20 | 0.98 | 0.49 | 0.08 | 0.49 |
| top-1 | Orders | 1.00 | 0.00 | 0.00 | 1.00 | 1.00 | 0.98 | 0.99 | 0.99 | 0.95 |
| | Payment | 0.99 | 0.40 | 0.00 | 1.00 | 0.93 | 0.91 | 0.97 | 0.98 | 1.00 |
| | User | 0.98 | 0.40 | 0.00 | 1.00 | 1.00 | 0.76 | 0.91 | 0.98 | 0.97 |
| | Avg. | 0.79 | 0.24 | 0.00 | 1.00 | 0.74 | 0.93 | 0.87 | 0.78 | 0.75 |
| | Carts | 1.00 | 0.60 | 0.40 | 1.00 | 0.76 | 1.00 | 1.00 | 1.00 | 1.00 |
| | Catalogue | 0.98 | 0.25 | 0.30 | 1.00 | 0.46 | 0.99 | 0.73 | 0.64 | 1.00 |
| top-3 | Orders | 1.00 | 0.00 | 0.09 | 1.00 | 0.96 | 0.99 | 0.99 | 0.99 | 1.00 |
| | Payment | 1.00 | 0.40 | 0.10 | 1.00 | 0.98 | 0.91 | 1.00 | 1.00 | 1.00 |
| | User | 1.00 | 0.62 | 0.11 | 1.00 | 1.00 | 0.78 | 0.94 | 1.00 | 1.00 |
| | Avg. | 1.00 | 0.37 | 0.20 | 1.00 | 0.83 | 0.93 | 0.93 | 0.93 | 1.00 |
| | Carts | 1.00 | 0.80 | 0.66 | 1.00 | 0.77 | 1.00 | 1.00 | 1.00 | 1.00 |
| | Catalogue | 0.99 | 0.52 | 0.60 | 1.00 | 0.49 | 1.00 | 0.85 | 0.80 | 1.00 |
| top-5 | Orders | 1.00 | 0.00 | 0.16 | 1.00 | 1.00 | 0.99 | 0.99 | 1.00 | 1.00 |
| | Payment | 1.00 | 0.40 | 0.19 | 1.00 | 0.96 | 0.91 | 1.00 | 1.00 | 1.00 |
| | User | 1.00 | 0.67 | 0.26 | 1.00 | 1.00 | 0.87 | 0.99 | 1.00 | 1.00 |
| | Avg. | 1.00 | 0.48 | 0.37 | 1.00 | 0.84 | 0.95 | 0.97 | 1.00 | 1.00 |

Table 2: The table presents the top-$l$ accuracy of various baselines on data collected from the Sock-shop application after injecting a memory leak failure into a specific microservice.

For the Sock-shop scenario, we considered state-of-the-art RCA baselines, including causal-RCA (Xin et al., 2023) (shown as cRCA), RUN (Lin et al., 2024), BARO (Pham et al., 2024), and RCD. Since RCG requires a causal graph, we used $\mathcal{C}$-PC, with postfixes indicating how $\mathcal{C}$ was chosen. For example, RCG-$k$ refers to using all conditioning sets up to size $k$, where $k \in \{0, 1\}$. We also introduced RCG-C, which avoids certain conditioning sets that may lead to faithfulness violations due to large support and finite samples. Additionally, we constructed a causal graph based on the system's call graph, denoted as RCG-Expert, to leverage expert knowledge for root cause identification.

Table 1 and 2 compares the top-$l$ accuracy of RCG with different baselines on the Sock Shop dataset. The results align with those from the synthetic data experiments in Section 6. Notably, RCD and RCG-0 perform similarly because, with $k = 0$, $\mathcal{C}$-PC is limited to marginal CI tests, producing a dense $\mathcal{C}$-essential graph. This results in more possible parent nodes, forcing RCG to condition on more variables, which can obscure the true root cause. However, when $k = 1$, RCG outperforms RCD by allowing $\mathcal{C}$-PC to use separating sets of size one. Increasing $k$ improves graph learning but demands larger sample sizes for reliable CI tests. RCG-C strikes a balance by refining the graph after $k = 0$, selectively conditioning on nodes with fewer states. Additionally, the system call graph shows that when a high-quality causal graph can be learned from observational data, RCG achieves strong target identification accuracy.

BARO achieved high top-1 accuracy on the Sock-shop data for both failure types. However, it disregards the correct causal order in the data-generating mechanism and is limited to continuous data.

| Outage | Nodes | Normal Samples | Failure Samples | Duration (hours) |
|--------|-------|----------------|-----------------|------------------|
| A | 152 | 4783 | 918 | 15 |
| B | 141 | 4626 | 1217 | 20 |
| C | 149 | 3464 | 110 | 2 |
| D | 146 | 7165 | 567 | 5 |

| Outage | RCG-0 | RCG-1 | RCG-2 | MI | BARO |
|--------|-------|-------|-------|-----|------|
| A | 7 | - | - | - | 9 |
| B | 1 | 6 | - | 9 | 6 |
| C | 1 | 1 | 1 | 1 | 8 |
| D | 5 | 5 | 6 | 3 | - |

Table 3: (**Left**) Summary of outages from a real-world production application. (**Right**) Rank of the root cause among the top 10 nodes for each baseline, with a rank of 1 indicating the highest-ranked node and a dash indicating the root cause was not found. RCG consistently outperforms MI and BARO at $k = 0$, but higher values of $k$ lead to a less reliable causal graph and decreased consistency. MI and BARO underperform by disregarding ancestral relationships and focusing solely on individual node changes.

As shown in Appendix J.3, BARO yields suboptimal results even with continuous data compared to RCG, which utilizes correct causal knowledge.

**Real Datasets.**

To assess the effectiveness of RCG and competing approaches, we collected data from a real-world production application over a seven-month period (January to July 2024), during which four outages were reported. For each incident, Software Reliability Engineers (SREs) documented key details, including outage duration, detection time, resolution method, and root cause. A summary of these outages is shown in the left table of Table 3. To identify the root cause, we presented the SREs with the top 10 ranked nodes from each baseline and asked them to confirm if the true root cause was among them. We report the rank of the root cause for each incident, where a lower rank indicates better performance by the method.

The right table in Table 3 compares the performance of MI, BARO, and RCG on the real-world dataset. The results show that RCG consistently outperforms both MI and BARO. With $k = 0$, RCG ranked the root cause within the top 5 nodes in three out of four cases. In contrast, BARO often ranked the root cause near the bottom and failed to identify it entirely in one case, highlighting the limitations of methods focused solely on detecting noticeable changes. This also underscores the drawbacks of relying on a single point of the distribution (such as the median), which may not accurately capture the shift between the two distributions. MI ranked the root cause in the top 3 for two outages but missed it in one case, likely due to the causal structure resembling a tree, which MI handles well due to data processing inequality. We also compared RCG at $k = 0$ and $k = 1$, finding that increasing $k$ did not consistently improve accuracy. In some cases, accuracy declined due to less reliable CI tests with larger separating sets, leading to incorrect parent node conditioning and inaccurate rankings.

In real-world applications, finding the right balance between accuracy and the informativeness of CI tests can be challenging. To address this, we propose an approach, $RCG^\star$, which combines results from different values of $k$ instead of selecting a single one. For example, one could take the top nodes from RCG-0 and combine them with the top nodes from RCG-1 until reaching a total of $l$ nodes. However, we leave the exploration of this combined approach for future work.

## 8 Conclusion

Identifying the root cause of system failures is a critical challenge in software systems. We argue that leveraging the causal structure of a system can provide valuable insights for diagnosing failures. We first demonstrate the value of the causal graph by showing that it can significantly reduce the number of invariance tests required. We show the lower bound on the number of marginal CI tests required to identify the root cause given the correct causal graph for any algorithm that uses solely marginal invariance tests. We then argue that the system's normal operational time can be leveraged to learn a partial causal graph. Based on this, we introduce an algorithm that systematically uses the partial causal graph to identify the root cause with a linear number of invariance tests. Empirical results show that our approach outperforms state-of-the-art methods, improving detectability.

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

## A   GRAPH NOTATIONS

**Definition A.1.** A graph $D = (\mathbf{V}, \mathbf{E})$ consists of a set of nodes (variables) $\mathbf{V}$ and a set of edges $\mathbb{E}$. We use $(X, Y)$ to denote an edge between a variable $X$ and another variable $Y$ in $D$. We consider graphs that contain only directed ($\rightarrow$) and undirected ($-$) edges. A directed graph has only directed edges. A partially directed graph may have both undirected and directed edges. A graph $D' = (\mathbf{V}', \mathbf{E}')$ is a *subgraph* of $D = (\mathbf{V}, \mathbf{E})$ and $D$ is a *supergraph* of $D'$ if $\mathbf{V}' \subseteq \mathbf{V}$ and $\mathbf{E}' \subseteq \mathbf{E}$. $D'$ is an *induced subgraph* of $D$ if $\mathbf{E}'$ are all edges in $\mathbf{E}$ between nodes in $\mathbf{V}'$.

**Definition A.2** (Path). Two vertices in a graph are said to be *adjacent* if there is an edge between them. Given a partially directed graph $D$, a *path* from $V_0$ to $V_n$ in $D$ is a sequence of distinct vertices $\langle V_0, V_1, \ldots, V_n \rangle$ such that for $0 \le i \le n-1$, $V_i$ and $V_{i+1}$ are adjacent. It is called a *causal* (or *directed*) path from $V_0$ to $V_n$ in $D$ if $V_i$ is a parent of $V_{i+1}$ for $0 \le i \le n-1$.

**Definition A.3** (Colliders). A consecutive triple of nodes $\langle X, Y, Z \rangle$ on a path is called a *collider* if both the edge between $X$ and $Y$ and the edge between $Y$ and $Z$ have arrowheads pointing to $Y$. If additionally $X$ and $Z$ are not adjacent, it is called *unshielded collider*. Any other consecutive triple is called a *non-collider*. If additionally, the two end vertices of the triple are not adjacent, it is called a *unshielded non-collider*.

**Definition A.4** (Ancestrality). In a graph $D$, for any two nodes $X, Y$ in $D$, if there is a directed edge $X \rightarrow Y$, then $X$ is a *parent* of $Y$ and $Y$ is a *child* of $X$ in $D$. If there is a causal path from $X$ to $Y$, then $X$ is called an *ancestor* of $Y$ and $Y$ is called a *descendant* of $X$. We denote a set of parents of $X$, a set of children of $X$, a set of ancestors of $X$, a set of descendants of $X$ and a set of non-descendants of $X$ in $D$ as $Pa_D(X), Ch_D(X), An_D(X), De_D(X)$ and $NDe_D(X)$ respectively. By convention, $X$ is both an ancestor and a descendant of $X$ in $D$. $X$ is called a *possible parent* of $Y$, denoted as $PossPa_D(X)$, if any of the following edges is in $D$: $\{X - Y, Xo \rightarrow Y, X \rightarrow Y, Xo—oY\}$. A source (or root) node has no parents. A sink node does not have any child.

In general, constraint-based algorithms can only learn up to an equivalence class of models, a set of DAGs that induce the same conditional independencies via d-separation, which gives the following definition.

**Definition A.5** (Markov Equivalence). Two DAGs $D_1, D_2$ with the same set of vertices are *Markov equivalent* if for any three disjoint set of vertices $\mathbf{X}, \mathbf{Y}, \mathbf{Z}$, $\mathbf{X}$ and $\mathbf{Y}$ are d-separated by $\mathbf{Z}$ in $D_1$ if and only if $\mathbf{X}$ and $\mathbf{Y}$ are d-separated by $\mathbf{Z}$ in $D_2$. A set of DAGs that encode the same set of conditional independence induced only by the causal Markov assumption is called the *Markov equivalence class*. Denote the Markov equivalence class of a DAG $D$ by $[D]$.

**Definition A.6.** (Essential Graph) The *essential graph* of a DAG $D$ has the same skeleton as $D$, with directed edges $X_i \rightarrow X_j$ if such edge direction between $X_i$ and $X_j$ holds for all DAGs in $[D]$, and undirected edges otherwise.

The essential graph is also called the *completed partially directed acyclic graph* (CPDAG) (Perković et al., 2017; Castelletti et al., 2018). Lee et al. (2024) defines the following set to restrict the conditioning sets used by all CI tests and the corresponding Markov equivalence class.

**Definition A.7** (Conditionally Closed Sets). For a DAG $D = (\mathbf{V}, \mathbf{E})$, let $\mathcal{I} = \{I_i\}$ be a set of CI statements of the form $I_i = (X, \mathbf{Z}, Y), i.e., (X \perp\!\!\!\perp Y | \mathbf{Z})$ or $(X \not\!\perp\!\!\!\perp Y | \mathbf{Z})$, where $X, Y \in \mathbf{V}, \mathbf{Z} \subset \mathbf{V}$. A set $\mathcal{C}$ is called *conditionally closed* if the following holds

1. $\emptyset \in \mathcal{C}$ and

2. $\exists X, Y \in \mathbf{V}, (X, \mathbf{C}, Y) \in \mathcal{I} \Rightarrow (A, \mathbf{C}, B) \in \mathcal{I}$ for all $A, B \in \mathbf{V}$ and for all $\mathbf{C} \in \mathcal{C}$

Generally, a DAG is only identifiable up to its Markov equivalence class since different DAGs can generate the same observational distributions. Here, an equivalence class of DAGs learned based on conditional independence relations restricted to $\mathcal{C}$ is defined as follows.

**Definition A.8** ($\mathcal{C}$-Markov equivalence). Two DAGs $D_1, D_2$ are *$\mathcal{C}$-Markov equivalent* if for any three disjoint subsets $\mathbf{X} \subset \mathbf{V}, \mathbf{Y} \subset \mathbf{V}, \mathbf{Z} \in \mathcal{C}$, $\mathbf{X}$ and $\mathbf{Y}$ are d-separated by $\mathbf{Z}$ in $D_1$ if and only if $\mathbf{X}$ and $\mathbf{Y}$ are d-separated by $\mathbf{Z}$ in $D_2$, where $\mathcal{C}$ is conditionally closed. The set of DAGs that encode the same set of conditional independence induced only by the causal Markov assumption with conditioning sets from $\mathcal{C}$ is called the *$\mathcal{C}$-Markov equivalence class*. We denote two DAGs $D_1, D_2$ that are $\mathcal{C}$-Markov equivalent as $D_1 \sim_\mathcal{C} D_2$.

Lee et al. (2024) defines a graphical representation that characterizes the set of d-separation relations based on $\mathcal{C}$ via the notion called $\mathcal{C}$-closure.

**Definition A.9** ($\mathcal{C}$-covered). Given a DAG $D = (\mathbf{V}, \mathbf{E})$ and a conditionally closed set $\mathcal{C}$, a pair of variables $X, Y$ is said to be $\mathcal{C}$-covered if there exists no separating set $\mathbf{C}$ in $\mathcal{C}$ to d-separate $X$ and $Y$ in $D$, i.e., $\nexists \mathbf{C} \in \mathcal{C}$ s.t. $(X \perp\!\!\!\perp Y | \mathbf{C})_D$.

**Definition A.10** ($\mathcal{C}$-closure). For a DAG $D$ and a conditionally closed set $\mathcal{C}$, the $\mathcal{C}$-closure of $D$, denoted as $\mathcal{S}_\mathcal{C}(D)$, is a graph that has the following properties:

1. If: $X, Y$ are $\mathcal{C}$-covered in $D$
   (i) if $X \in An_D(Y)$, then $X \to Y$ in $\mathcal{S}_\mathcal{C}(D)$, (ii) if $Y \in An_D(X)$, then $Y \to X$ in $\mathcal{S}_\mathcal{C}(D)$, (iii) else $X \leftrightarrow Y$ in $\mathcal{S}_\mathcal{C}(D)$.

2. Else: $X, Y$ are not adjacent in $\mathcal{S}_\mathcal{C}(D)$.

The relationship between a DAG and $\mathcal{C}$-closure graph is described by the following lemma, which says that all d-separation relations based on $\mathcal{C}$ hold in a DAG also hold $\mathcal{C}$-closure.

**Lemma A.11.** *Lee et al. (2024) $\mathcal{C}$-closure graph $\mathcal{S}_\mathcal{C}(D)$ of a DAG $D$ entails the same d-separation statements conditioned any $\mathbf{C} \in \mathcal{C}$ as the DAG, i.e., $(X \perp\!\!\!\perp Y | \mathbf{C})_D \Leftrightarrow (X \perp\!\!\!\perp Y | \mathbf{C})_{\mathcal{S}_\mathcal{C}(D)}, \forall \mathbf{C} \in \mathcal{C}$.*

**Theorem A.12.** *Lee et al. (2024) Two DAGs $D_1, D_2$ are $\mathcal{C}$-Markov equivalent if and only if $\mathcal{S}_\mathcal{C}(D_1)$ and $\mathcal{S}_\mathcal{C}(D_2)$ are Markov equivalent.*

The representation of a set of Markov equivalence classes of $\mathcal{C}$-closure graphs is called the $\mathcal{C}$-essential graph.

**Definition A.13.** [edge unions: $—, o{—}o, o{\to}$] The edge union operations of a set of $\mathcal{C}$-closure graphs are defined as: (i) $X — Y := X \to Y \cup X \leftarrow Y$, (ii) $X\, o{—}o\, Y := X \to Y \cup X \leftarrow Y \cup X \leftrightarrow Y$, (iii) $X\, o{\to}\, Y := X \to Y \cup X \leftrightarrow Y$. We use $*$ to denote a wildcard mark of any of the following marks: a tail, an arrowhead, and a circle.

**Definition A.14** ($\mathcal{C}$-essential graph). For any DAG $D$, the edge union of all $\mathcal{C}$-closure graphs that are Markov equivalent to $\mathcal{S}_\mathcal{C}(D)$ is called the $\mathcal{C}$-essential graph of $D$, denoted as $\varepsilon_\mathcal{C}(D)$.

Note that $\mathcal{C}$-essential graph is a supergraph of the essential graph. The more conditioning sets that are included in $\mathcal{C}$, the closer that the $\mathcal{C}$-essential graph will be like the essential graph. For learning $D_{aug}$, we need to leverage distributional invariances across the normal and anomalous datasets via the following two assumptions. For a more detailed discussion on these assumptions, please see Jaber et al. (2020).

**Assumption A.15** ($\Psi$-Markov conditions). Let $\mathbf{P}$ denote an ordered tuple of distributions and let $\mathcal{I}$ be an ordered tuple of the children of F-NODE. $\mathbf{P}$ is called *$\Psi$-Markov* relative to a graph $D_{aug} = (\mathbf{V}, \mathbf{E})$ if the following holds for $\mathbf{Y}, \mathbf{Z}, \mathbf{W} \subseteq \mathbf{V}$:

1. For $\mathbf{I}_i \in \mathcal{I}$: $P_i(\mathbf{y}|\mathbf{w}, \mathbf{z}) = P_i(\mathbf{y}|\mathbf{w})$ if $\mathbf{Y} \perp\!\!\!\perp \mathbf{Z}|\mathbf{W}$ in $D_{aug}$

2. For $\mathbf{I}_i, \mathbf{I}_j \in \mathcal{I}$: $P_i(\mathbf{y}|\mathbf{w}) = P_j(\mathbf{y}|\mathbf{w})$ if $\mathbf{Y} \perp\!\!\!\perp \mathbf{K}|\mathbf{W_K}$ in $D_{aug}{}_{\underline{\mathbf{W_K}}, \overline{\mathbf{R(W)}}}$

, where $\mathbf{K} := (\mathbf{I_i} \setminus \mathbf{I_j}) \cup (\mathbf{I_j} \setminus \mathbf{I_i})$, $\mathbf{W_K} := \mathbf{W} \cap \mathbf{K}$, $\mathbf{R} := \mathbf{K} \setminus \mathbf{W_K}$, and $\mathbf{R(W)} \subseteq \mathbf{R}$ are non-ancestors of $\mathbf{W}$ in $D_{aug}$.

**Assumption A.16** (c-faithfulness). A tuple of distributions $\mathbf{P}$ are said to be *c-faithful* to $D_{aug}$ if the converse of each of the $\Psi$-Markov conditions holds.

## B  RELATED WORK

**Root Cause Analysis in Microservices.** Root Cause Analysis (RCA) is done both online (Wang et al., 2023a) and offline (Deng & Hooi, 2021), often relying on system dependency graphs (Chen et al., 2014). Previous approaches have used statistical techniques, deep neural networks, and graph representation (Brandón et al., 2020; Capozzoli et al., 2015; Ma et al., 2020). For instance, (Lin et al., 2018) uses z-scores to compare the distributions of normal operation and anomalous system data. The method finds the root cause by identifying nodes that deviate the most between two distributions, but it imposes normality assumptions on the data and it is sensitive to outliers. Li et al. (2022) also uses similar techniques with a call graph provided by expert knowledge to adjust the scores. Pham et al. (2024) improves this idea by using median and interquartile range instead, but the method is not applicable to discrete distributions. Wang et al. (2023b) used both individual and topological time series data to capture interdependencies between microservices, while Xin et al. (2023) introduced a gradient-based causal structure learning method to generate weighted causal graphs and developed a root cause inference method called CausalRCA. Recently, Lin et al. (2024) proposed RUN, a method that forecasts time series by constructing a neural network for each system metric and then uses the forecasted data to build a Granger causal graph. During the diagnosis stage, RUN, like other algorithms, applies a weighted personalized PageRank algorithm to traverse the graph and identify the root cause. A closely related work to ours is RCD (Ikram et al., 2022), where Ikram et al. (2022) presented a causal framework that treats failure as an intervention. They developed a hierarchical approach to causal discovery by randomly partitioning the set of observed variables and using a series CI tests in each partition to produce a set of potential root causes. This approach is particularly relevant to our work, as it also employs CI tests to localize and pinpoint the failure's root cause. However, despite the innovative contributions of these recent studies, we argue that a critical aspect has been overlooked: the opportunity to utilize normal operation periods to develop a more efficient and effective RCA method for failure periods.

**Causal Discovery with Bounded Conditioning Set Size.** Given that the use of CI tests is a central aspect of our work, we provide a brief overview of recent advances in causal discovery, particularly those focused on bounding the size of CI tests. Causal discovery often relies on a series of CI tests to determine relationships between variables. However, this approach can be problematic, as the statistical power of CI tests diminishes with a finite sample size or when the conditioning set is large (Shah & Peters, 2020). A promising direction in addressing this issue has been the exploration of methods to restrict the size of the conditioning set. In the absence of latent confounders, Wienöbst & Liskiewicz (2020) introduced a sound and complete algorithm known as Low-Order Causal Inference (LOCI), which learns a graphical representation based on CI relations of order $k$ or lower. Similarly, Kocaoglu (2023) provided a novel characterization of the graphical representation termed the $k$-essential graph, along with a sound learning algorithm to construct it. Building on these ideas, Lee et al. (2024) proposed an approach that further restricts the conditioning sets for all CI tests so long these tests include all marginal tests. Our objective in this work is to integrate these recent advancements to develop and utilize a more robust causal graph than the current state-of-the-art in RCA literature.

## C  THEOREMS AND PROOFS

For the sack clarity, we first provide the Theorem 1 from Shangqi et al. (2023) and Theorem 2 from Tao et al. (2019). Shangqi et al. (2023) term the IGS problem as the POMS problem and they refer to a DAG as an input graph.

**Theorem C.1** (Shangqi et al. (2023)). *For the POMS problem, let $n$ represent the number of vertices in the input graph $D$ and $d$ denote the maximum vertex out-degree in $D$. Both of the following statements are true:*

- *There is an algorithm that can find the target in $O(\log_{1+k} n + (d/k) \log_{1+d} n)$ probs.*

- *Any POMS algorithm must perform $\Omega(\log_{1+k} n + (d/k) \log_{1+d} n)$ probs to find the target in the worse case.*

**Theorem C.2** (Tao et al. (2019)). *Both of the following statements are true about the IGS problem:*

- `DFS-interleave` *asks at most $\lceil \log_2 h \rceil \cdot (1 + \lfloor \log_2 n \rfloor) + (d - 1) \cdot \lceil \log_d h \rceil$ questions.*

- *Any algorithm must ask at least $(d - 1) \cdot \lceil \log_d h \rceil$ questions in the worst case.*

We provide the pseudocode of `DFS-interleave`, which has been modifed for RCA, in Algorithm 7.

**Lemma 4.1.** *Given a causal graph $D$, if $(F \perp\!\!\!\perp X)_P$ for some $X \in \mathbf{V}$, then $A \notin Ch_{D_{aug}}(F)$ for all $A \in An_D(X)$, where $P$ is any joint distribution between variables on $D_{aug}$.*

*Proof.* For the sake of contradiction, suppose $F \to A$ in $D_{aug}$ for some $A \in An_D(X)$. Since $A$ is an ancestor of $X$ in $D$, there must be a directed path $q$ from $A$ to $X$ in $D$. Thus, $q$ must also exist in $D_{aug}$. Consider the path obtained by concatenating $F \to A$ with $q$ in $D_{aug}$. This path must be d-connecting in $D_{aug}$. Thus, it must be that $(F \not\perp\!\!\!\perp X)_{D_{aug}}$. From interventional faithfulness, we have that $(F \not\perp\!\!\!\perp X)_P$, which is a contradiction. $\square$

**Lemma 4.2.** *Given a causal graph $D$, if $(F \not\perp\!\!\!\perp X)_P$ for some $X \in \mathbf{V}$, then then $Q \notin Ch_{D_{aug}}(F)$ for all $Q \in NAn_D(X)$, where $P$ is any joint distribution between variables on $D_{aug}$.*

*Proof.* For the sake of contradiction, suppose $F \to Q$ in $D_{aug}$ for some $Q \in NAn_D(X)$. Since $Q$ is a non-ancestor of $X$ in $D$, without loss of generality, there are several cases: (i) there exists a directed path $q$ from $X$ to $Q$ in $G$ (ii) there is no path between $Q$ and $X$ in $D$ and (iii) any path $p$ between $X$ and $Q$ must have a collider on $p$ in $D$.

For case (i), $q$ must also exist and be directed in $D$. By concatenating the path from $X$ to $Q$ and $F \to Q$, we see the path from $F$ to $X$ is blocked. Thus, we have $(F \perp\!\!\!\perp X)_D$, which implies $(F \perp\!\!\!\perp X)_P$ by Assumption A.15, which is a contradiction.

For case (ii), there is no path between $X$ and $Q$ in $D$, which implies $(F \perp\!\!\!\perp X)_D$ so that we reach the same contradiction.

For case (iii), every collider on any path $p$ between $Q$ and $X$ must also be in $D$ such that we have $(F \perp\!\!\!\perp X)_D$ by concatenating $F \to Q$ with $p$, which implies $(F \perp\!\!\!\perp X)_P$ by Assumption A.15, which is a contradiction. $\square$

**Lemma 4.3.** *Consider a DAG $D = (\mathbf{V}, \mathbf{E})$ with a single sink node and $D'$ be a DAG by reversing every edge direction in $\mathbf{E}$, let $Q(X)$ be a query to the oracle on whether some $X \in \mathbf{V}$ has a directed path to an unknown target node $R \in \mathbf{V}$.*

$$Q(X) = yes \Leftrightarrow (F \not\perp\!\!\!\perp X)_P \tag{1}$$

*. Therefore, if $Q(X) = yes$, then $X \in An_{D'}(R)$. If $Q(X) = no$, then $X \in NA_{D'}(R)$.*

*Proof.* Consider some nodes $X \in \mathbf{V}$, suppose $(F \perp\!\!\!\perp X)_P$, then $X \in NDe_D(R)$ by Lemma 4.1. Note that $NDe_D(R) = NAn_{D'}(R)$ due to $De_D(R) = An_{D'}(R)$ by the given conditions for $D$ and $D'$. Therefore, $X \in NAn_{D'}(R)$. As $NAn_{D'}(R) \Leftrightarrow Q(X)=$ no. We have that $(F \perp\!\!\!\perp X)_P \Rightarrow Q(X) = $ no. Similarly, suppose $(F \not\perp\!\!\!\perp X)_P$, then $X \in De_D(R)$ by Lemma 4.2. As $De_D(R) = An_{D'}(R)$, we have that $(F \not\perp\!\!\!\perp X)_P \Rightarrow X \in An_{D'}(R)$, which is equivalent to $Q(X) = $ yes. $\square$

**Theorem 4.4.** *Given a causal graph $D$ with a single sink node, any algorithm the only uses marginal invariance tests must perform $\Omega(\log_2 n + d \log_{1+d} n)$ many tests to find the single root cause in the worst case, where $d$ is the maximum in-degree of $D$ and $n$ is the number of nodes. There exists an algorithm that finds the root cause with $\mathcal{O}(\log_2 n + d \log_{1+d} n)$ marginal invariance tests.*

*Proof.* This follows from Lemma 4.3 and Theorem 1 in (Shangqi et al., 2023), which says that any algorithm must ask $\Omega(\log_2 n + d \log_{1+d} n)$ queries to identify the target node selected by an adversary in a DAG $D'$ with a single root node for the problem of IGS, where $d$ is the maximum out-degree in $D'$ and there is an algorithm that can find the target node in $\mathcal{O}(\log_2 n + d \log_{1+d} n)$ number of queries. □

The following lemma is similar to Lemma 1 in Wienöbst & Liskiewicz (2020) but its setup is based on CIs restricted to the conditionally closed set $\mathcal{C}$.

**Lemma 5.2.** *Given a distribution $P$ defined over a set of CIs based on a conditionally closed set $\mathcal{C}$, for any $X, Y \in \mathbf{V}$ and $\mathbf{Z} \in \mathcal{C}$, if $(X \perp\!\!\!\perp Y \,|\mathbf{Z})_P, (X \not\perp\!\!\!\perp W \,|\mathbf{Z})_P$, then no DAG faithful to $P$ contains the edge $W \to Y$.*

*Proof.* For the sake of contradiction, assume that there is DAG that contains the directed edge from $W$ to $Y$. Since $(X \not\perp\!\!\!\perp W \,|\mathbf{Z})_P$, we have that $X$ is d-connecting with $W$ given $\mathbf{Z}$, concatenating this d-connecting path with $W \to Y$, we have that $X$ is also d-connecting with $W$ given $\mathbf{Z}$, which is a contradiction. □

**Lemma 5.3.** *Let $M$ be the graph returned by Algorithm 2, $F$ is not adjacent to $X$ in $D_{aug}$ if and only if $F$ is d-separated with $X$ given $PossPa_M(X)$ in $D_{aug}$.*

*Proof.* We will prove the if ($\Rightarrow$) direction.

We first give a critical insight. We note that if F-NODE points to any variable that is a collider $H$ on some paths $p$ in $D_{aug}$, then running marginal tests must have allowed us to orient $Fo\to H \leftarrow oU$ and $Fo\to H \leftarrow oQ$ for some variables $U, Q$ on $p$ in the given $\varepsilon_{\mathcal{C}}(D)$ due to Lemma 5.2. Thus, we call this resulting graph $M$ rather than $\varepsilon_{\mathcal{C}}(D)$. If $F$ is marginally independent with all members in the adjacency set of $H$, then the result follows.

Suppose there is more than one node being marginally dependent on $F$. We call this set $\mathbf{Z}$. Then, we know $F$ must have a directed path to all such nodes $Z \in \mathbf{Z}$ in $D_{aug}$ as there is no incoming edges to $F$ and each of these nodes is marginally dependent with $F$. We will prove the claim that if $F$ is not adjacent to $Z$ in $D_{aug}$, then $F$ is d-separated with $Z$ given $PossPa_M(Z)$ in $D_{aug}$ for all $Z \in \mathbf{Z}$.

For the sake of contradiction, assume that $F$ is d-connecting with $Z$ given $PossPa_M(Z)$ in $D_{aug}$. First, we note that $PossPa_M(Z)$ must contain all parents of $Z$ in $D_{aug}$. Since there exists a directed path from $F$ to $Z$, we call this path $r$ as shown below:

$$F \to T \to \ldots \to W \to \ldots \to Z. \tag{4}$$

Then, since $PossPa_M(Z)$ must contain all parents of $Z$, we consider two cases: (i) there exists a backdoor active path from $F$ to $Z$ by concatenating with a subpath of $r$ as follows:

$$F \to T \to \ldots \to W \leftarrow Q \to \ldots \to Z \tag{5}$$

and case (ii): there exists a d-connecting path from $F$ to $Z$ given some variables $K$ as follows

$$F \to T \to \ldots \to W \to \ldots \to K \leftarrow Z \tag{6}$$

Case (i) - there exists a backdoor active path from $F$ to $Z$ by concatenating with a subpath of $r$: We will first show a contradiction in case (i). Note that we cannot have $Q \in An_{D_{aug}}(Z)$. To see that, suppose $Q$ and $Z$ is $\mathcal{C}$-covered, then $Q$ must be in $PossPa_M(Z)$ as $(F \perp\!\!\!\perp Z)_P$ so that Algorithm 2 will not change the orientation of this edge. Suppose they are not $\mathcal{C}$-covered, there exists a member along this path from $Q$ to $Z$ conditioned on which d-separates $Q$ and $Z$, which contradicts with the fact there is an active backdoor path. Thus, there exists a collider $U_1$ on the path from $Q$ to $Z$ as follows.

$$F \to T \to \ldots \to W \leftarrow Q \to \ldots \to U_1 \leftarrow \ldots Z \tag{7}$$

Then, a member in $De_{D_{aug}}(U_1)$ must be in $PossPa_M(Z)$ in order for the path in (7) to be a d-connecting path from $F$ to $Z$. Consider $U_1$ is a child of $Z$ in $D_{aug}$ and the node $U_2$ that is closest to $U_1$ to form $U_2 \to U_1 \leftarrow Z$ in $D_{aug}$. If $U_2$ and $Z$ are not $\mathcal{C}$-covered, then $\langle U_2, U_1, Z \rangle$ must be unshielded in $M$. Then, $U_1$ cannot be in $PossPa_M(Z)$ as $Z*\!\to U_1$ must have been oriented as an

unshielded collider in $M$, which is a contradiction. If $U_2$ and $Z$ are $\mathcal{C}$-covered, then $U_2$ is adjacent to $Z$ in $M$. We will consider two cases: (a) $U_2 \notin PossPa_M(Z)$ and (b) $U_2 \in PossPa_M(Z)$.

Case(a): $U_2 \notin PossPa_M(Z)$: Suppose $U_2 \notin PossPa_M(Z)$, then it must be that $U_2 \leftarrow* Z$ in $M$. Then, we have a collider $\langle U_3, U_2, Z \rangle$ on the path from $W$ to $Z$, where $U_3$ is the next closest node to $U_2$ on the same path. If $\langle U_3, U_2, Z \rangle$ is unshielded in $M$, then the $\mathcal{C}$-essential graph provided would have oriented $U_3 *\!\to U_2 *\!\to U_1$ in $M$ by using the first Meek rule. Then, using acyclicity (second Meek rule) infers that $Z *\!\to U_1$ in $M$ such that $U_1 \notin PossPa_M(Z)$. Since we have $Z *\!\to U_2$ in $M$, there exists a $\mathcal{C}$-closure graph $\mathcal{S}_\mathcal{C}(D')$ of some causal graph $D'$ that is $\mathcal{C}$-Markov equivalent to $D_{aug}$ by Theorem A.12 and $Z \in An_{D'}(U_2)$. The path $F \to W \leftarrow Q \to U_2$ concatenating this directed path from $Z$ to $U_2$ cannot be a d-connecting path from $F$ to $Z$ given $PossPa_M(Z)$ because the child of $Z$ on this path would not be in $PossPa_M(Z)$ as $U_1$ is also a child of $Z$. Hence, we reach a contradiction. Suppose $\langle U_3, U_2, Z \rangle$ is shielded, we see that the same argument repeats by picking the next closest node to $U_3$ until we have reached that $\langle Q, U_j, Z \rangle$ is shielded for some $j$, if $Q \in PossPa_M(Z)$, then we will also reach a contradiction because the path in (7) will no longer be active from $F$ to $Z$ given $PossPa_M(Z)$. We will see that it is impossible to have $Q \notin PossPa_M(Z)$ either. Suppose $Q \notin PossPa_M(Z)$, then there must exist $Z *\!\to Q$ in $M$. However, this is also a contradiction for the following reason: any DAGs that is $\mathcal{C}$-Markov equivalent to $D_{aug}$ must have $F \to \ldots \to W \leftarrow Q$ as $F$ has a directed path to $W$ and no incoming edges. Having $Z *\!\to Q$ in $M$ implies, for some DAG $D''$, there exists a $\mathcal{C}$-closure graph $\mathcal{S}_\mathcal{C}(D'')$ that is Markov equivalent to $\mathcal{S}_\mathcal{C}(D_{aug})$ has $Z \to Q$. We see that there will be a directed cycle in $D''$ as $F$ must have a directed path to $Z$ and $F \to \ldots \to W \leftarrow Q$ and $Z$ has a directed path to $Q$.

Case(b): $U_2 \in PossPa_M(Z)$: Suppose $U_2 \in PossPa_M(Z)$. Consider the node that is closest to $U_2$ in the path in (7) from $Q$ to $Z$. We call this node $U_3$. Since $U_2 \in PossPa_M(Z)$, $\langle U_3, U_2, Z \rangle$ cannot be an unshielded collider on the path from $Q$ to $Z$ in $M$. That implies $\langle U_3, U_2, Z \rangle$ must be shielded. We can repeat this argument by picking the next closest node until the next closest node is $Q$ so that we have $\langle Q, U_j, Z \rangle$ being shielded for some $j$. Then, the same argument as in case (a) repeats, reaching a contradiction.

Case (ii): there exists a d-connecting path from $F$ to $Z$ given some variables $K$: Now, we consider the case (ii) with the path in (6). Consider the node closest to $K$. We call this node $K_1$ such that $\langle K_1, K, Z \rangle$ form a collider on the path in (6) in $D_{aug}$. If $\langle K_1, K, Z \rangle$ is unshielded in $M$, then $K$ cannot be in $PossPa_M(Z)$ as $Z \star \to K$ would have been oriented by $\mathcal{C}$-PC. Suppose $\langle K_1, K, Z \rangle$ is not unshielded in $M$. Consider the node closest to $K_1$. We call this node $K_2$ If $\langle K_2, K_2, Z \rangle$ is unshielded in $M$, then $K$ cannot be in $PossPa_M(Z)$ as $Z *\!\to K$ would have been oriented by $\mathcal{C}$-PC. We can see this repeated argument until the closest node to $K_i$ for some $i$ is $T$. Then, $T$ must be in $PossPa_M(Z)$. Therefore, $F$ is d-separated from $Z$ given $PossPa_M(Z)$, which is a contradiction, blocking the path from $W$ to $Z$ such that $F$ is d-separated from $Z$ given $PossPa_M(Z)$, which is a contradiction.

For the only if direction, for the sake of contradiction, assume $F$ and $X$ is adjacent in $D_{aug}$. Since $F$ and $X$ are d-separated given the possible parents set of $X$ in $M$, then there is no d-connecting path from $F$ to $X$ given the possible parents set of $X$, which is a contradiction as $F$ is adjacent to $X$. $\qquad\square$

**Corollary 5.4.** *Given two graphs $M_1, M_2$ returned by Algorithm 2 based on two different $\mathcal{C}$-essential graphs $\varepsilon_{\mathcal{C}_1}(D)$ and $\varepsilon_{\mathcal{C}_2}(D)$, if $\mathcal{C}_1 \subset \mathcal{C}_2$, then $|PossPa_{M_1}(X)| \geq |PossPa_{M_2}(X)|$.*

*Proof.* Since $\mathcal{C}_1 \subset \mathcal{C}_2$, $\mathcal{C}$-PC will conduct more CI tests based on $\mathcal{C}_2$, which can result in a sparser $\mathcal{C}$-essential graph, it follows that $|PossPa_{M_1}(X)| \geq |PossPa_{M_2}(X)|$ for all $X \in \mathbf{V}$. $\qquad\square$

## D  ALGORITHMS

---

**Algorithm 3** $\mathcal{C}$-PC Lee et al. (2024)

---

**input** Observational data $\mathbf{V}$, a conditionally closed set $\mathcal{C}$, CI tester
1: Initiate a complete graph $M$ among the set of observed variables with circle edge $o$—$o$.
2: Find separating sets $S_{X,Y}$ for every pair $X, Y \in V$ by conditioning on $C \in \mathcal{C}$.
3: Update $M$ by removing the edges between pairs that are separable.
4: Orient unshielded colliders of $M$: For any induced subgraph $Xo$—$oZo$—$oY$ or $Xo \rightarrow Zo$—$oY$ or $Xo$—$oZ \leftarrow oY$, set $Xo \rightarrow Z \leftarrow oY$ for any non-adjacent pair $X, Y$ where $S_{X,Y}$ does not contain $Z$.
5: $M \leftarrow$ **kPC_Orient**$(M)$
6: **return** $M$

---

**Algorithm 4** kPC_Orient Kocaoglu (2023)

---

**input** Mixed graph $M$
1: $M \leftarrow$ FCI_Orient$(M)$ {See Algorithm 5}
2: For any variable $X$ that has no incoming edges, construct the sets $\mathcal{B}, \mathcal{Q}$ :

$$\mathcal{B} = \{Y \in Ne(X) : Xo \rightarrow Y\},$$
$$\mathcal{Q} = \{Z \in Ne(X) : Xo\text{—}oZ\}$$

and define sets $\mathcal{B}^\star$ as the set of variables that are non-adjacent to any of the nodes in $\mathcal{Q}$ and $\mathcal{Q}^\star$ as the set of variables that are non-adjacent to other variables in $\mathcal{Q}$:

$$\mathcal{B}^\star = \{Y \in \mathcal{B} : Y, Z \text{ are non-adjacent } \forall Z \in \mathcal{Q}\},$$
$$\mathcal{Q}^\star = \{Z' \in \mathcal{Q} : Z', Z \text{ are non-adjacent } \forall Z' \neq Z, Z' \in \mathcal{Q}\}$$

3: $\mathcal{R}11$ : Orient $Xo \rightarrow Y$ as $X \rightarrow Y, \forall Y \in \mathcal{B}^\star$
4: $\mathcal{R}12$ : Orient $Xo$—$oY$ as $X$—$Y, \forall Z \in \mathcal{Q}^\star$
5: **return** $M$

---

**Algorithm 5** FCI_Orient Zhang (2008)

---

**input** Mixed graph $M$
1: Apply the orientation rules of $\mathcal{R}1, \mathcal{R}2, \mathcal{R}3$ of Zhang (2008) to $M$ until none applies.
2: Apply the orientation rules of $\mathcal{R}8, \mathcal{R}9, \mathcal{R}10$ of Zhang (2008)
3: **return** $M$

---

**Algorithm 6** CONSTRUCT-HEAVY-PATH-DFS-TREE Tao et al. (2019)

---

**input** DAG $D = (\mathbf{V}, \mathbf{E})$
**output** A heavy-path-DFS-tree $T$
1: Create a stack $\mathcal{S}$ with the root node $R$ in $D$ and mark $R$ visited.
2: **repeat**
3:    $J \leftarrow$ get the top member in the stack.
4:    **if** $J$ has any child $A$ that has not been visited previously **then**
5:       $A' \leftarrow$ Find the child that can reach the highest number of nodes that have not been visited via a directed path.
6:       Push $A'$ into the stack $\mathcal{S}$ and mark it visited.
7:    **else**
8:       Pop $J$ out of the stack $\mathcal{S}$.
9:    **end if**
10: **until** $\mathcal{S}$ is empty

---

---

**Algorithm 7** Modified IGS (DFS-Interleave Tao et al. (2019)) for RCA

---

**input** DAG $D = (\mathbf{V}, \mathbf{E})$, interventional data $\mathcal{D}$, CI tester,
**output** A root cause $R$
 1: **if** $D$ has more than one sink node **then**
 2:  $D \leftarrow$ Add a dummy vertex $S$ to $D$ where all the sink nodes in $D$ point to $S$.
 3: **end if**
 4: $D \leftarrow$ Reverse all the edges in $D$
 5: $T \leftarrow$ **CONSTRUCT-HEAVY-PATH-DFS-TREE**$(D)$ {See Algorithm 6}
 6: $\hat{R} \leftarrow$ Select the root of $T$
 7: **repeat**
 8:  $\pi \leftarrow$ Select the leftmost $\hat{R}$-to-leaf path of $T$
 9:  $U \leftarrow$ Perform binary search on $\pi$ to find the last node $U$ that gives $(F \not\perp\!\!\!\perp U)_P$.
10:  $W \leftarrow$ Find the leftmost child of $U$ in $T$ where $(F \not\perp\!\!\!\perp W)_P$.
11:  **if** $W$ does not exists **then**
12:    **return** $U$
13:  **else**
14:    **update** $\hat{R} \leftarrow W$
15:  **end if**
16: **until** $\hat{R}$ has not been updated.

---

# E SAMPLE RUN OF RCD IKRAM ET AL. (2022)

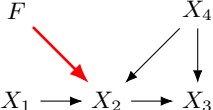

Figure 3: An example to show how RCD works. RCD would need increase the size of the separating set to 2 to find the root cause ($X_2$). However, we can leverage the causal graph to know precisely the separating set for every node.

RCD is based on the observation that a failure in a microservice can be treated as an intervention in the underlying causal graph. By treating the root cause as the interventional target, RCD leverages recent advances in causal discovery to identify the root cause. Consistent with the broader causal discovery literature, RCD determines the interventional target (the root cause) through a series of CI tests. RCD operates by introducing a special node, referred to as F, into the dataset and connecting it to every other node in a complete undirected graph. The algorithm's primary goal is to trim down the children of F, as the true root cause will ultimately be the sole remaining child. However, due to the lack of information about the underlying graphical structure, RCD must condition on every possible set of variables until it identifies a separating set that can exclude a potential node as the root cause.

For instance, consider the ground truth causal graph shown in Figure 3, where the root cause is $X_2$. Initially, RCD constructs an undirected graph with F having outgoing edges to every node. It begins with a separating set of size 0 and executes all possible CI tests. After conducting the tests $(F \perp\!\!\!\perp X_1)_P$ and $(F \perp\!\!\!\perp X4)_P$, RCD removes the edges between F and both $X_1$ and $X_4$. At this point, only two candidates for the root cause remain: $X_2$ and $X_3$. To narrow it down to the true root cause, RCD increases the size of the separating set. If it tests $X_2$, it runs $(F \not\perp\!\!\!\perp X2|X_3)_P$. Since $X_2$ is the root cause, it cannot be independent of F. When testing $X_3$ by running $(F \not\perp\!\!\!\perp X3|X_2)_P$, conditioning on $X_2$ opens a backdoor path from F to $X_3$, preventing its elimination. RCD then increases the size of the separating set once more and runs $(F \perp\!\!\!\perp X3|X_2, X_4)_P$, which removes the edge between F and $X_3$. Finally, RCD stops, identifying $X_2$ as the root cause.

Since RCD lacks access to the causal graph, it must perform CI tests on all possible conditioning sets (up to size 2) to identify the root cause, resulting in an exponential growth in tests and higher computational costs. To address this, RCD limits the conditioning set size using a hyperparameter, though this can lead to incomplete results. We propose that knowing the causal graph can signif-

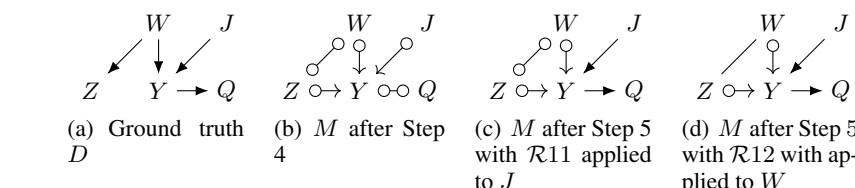

(a) Ground truth $D$

(b) $M$ after Step 4

(c) $M$ after Step 5 with $\mathcal{R}11$ applied to $J$

(d) $M$ after Step 5 with $\mathcal{R}12$ with applied to $W$

Figure 4: (a)-(d): Given $\mathcal{C} = \{\emptyset, \{Y\}\}$, this is an example of the execution of Algorithm 3. Particularly, 4(d) shows the output of $\mathcal{C}$-PC for learning the ground truth in 4(a).

icantly reduce the number of required CI tests. A causal graph provides precise separating sets, allowing the root cause to be identified with at most $n$ CI tests, where $n$ corresponds to the number needed for validation of the structure.

## F    SAMPLE RUN OF $\mathcal{C}$-PC ALGORITHM AND INTERPRETATIONS OF $\mathcal{C}$-ESSENTIAL GRAPH

As $\mathcal{C}$-PC is highly relevant to our algorithm. We give a sample run of $\mathcal{C}$-PC algorithm as in Lee et al. (2024) to demonstrate how it works in Figure 4. The ground truth is provided in Figure 4(a). Suppose we let $\mathcal{C} = \{\emptyset, \{Y\}\}$. It means that one will only conduct all marginal independence tests and CI tests with conditioning set $\{Y\}$. The resulting graphical representation after finishing step 4 of Algorithm 3 is in Figure 4(b). The definition of various marks on the graph is provided in Definition A.13. Then, by applying some orientation rules in step 5 of Algorithm 3, we can obtain the final output shown by Figure 4(d).

We will use the output of $\mathcal{C}$-PC in figure 4(d) to illustrate the meaning of a $\mathcal{C}$-essential graph. The interpretation of this graphical object known as $\mathcal{C}$-essential graph is that it represents a set of conditional independence relations induced by the ground truth in Figure 4(a) with respect to the set $\mathcal{C} = \{\emptyset, \{Y\}\}$. These CI relations are $(Z \perp\!\!\!\perp J)_P, (W \perp\!\!\!\perp J)_P, (Z \perp\!\!\!\perp Q|Y)_P, (Q \perp\!\!\!\perp J|Y)_P, (W \perp\!\!\!\perp Q|Y)_P, (Z \perp\!\!\!\perp Q|Y)_P$. Both the arrowheads and directed edges e.g. $J \to Y$ in Figure 4(d) are invariant across all the DAGs that are $\mathcal{C}$-Markov (see Definition A.8) to the ground truth by Lemma A.11 and Theorem A.12. An undirected edge $Z - W$ denotes that there exists a $\mathcal{C}$-closure graph that has $Z \to W$ and another $\mathcal{C}$-closure graph that has $W \to Z$ within the same Markov equivalence class. Please see Definition A.10 for the relationships between DAGs and $\mathcal{C}$-closure graphs. As $\mathcal{C}$-essential graph represents a set of $\mathcal{C}$-closure graphs, the edge union operation (see Definition A.13) is then used to represent different orientations in these $\mathcal{C}$-closure graphs that are Markov equivalent.

## G    AN EXAMPLE THAT SHOWS THE BENEFITS OF LEMMAS 4.1 AND 4.2

We will use Figure 5 to illustrate how Lemmas 4.1 and 4.2 may help identify the root cause, which is $X_1$ in this case, with less than $n$ invariance tests. We can start by arbitrarily picking a variable for testing conditional independence with $F$. Suppose we select $X_2$ to test whether $(F \perp\!\!\!\perp X_2)_P$. By Assumption A.16, we will observe $(F \not\perp\!\!\!\perp X_2)_P$. Then, Lemma 4.2 says that $X_3$ cannot be the root cause. Suppose we pick $X_1$ to test for conditional independence, then we will observe $(F \perp\!\!\!\perp X_1)_P$. Then, by Lemma 4.1, we know that $X_5$ cannot be the root cause either. Then, we are only left with $X_4$ to test for conditional independence. This results in a total of 3 marginal independence tests, which is less than $n = 5$.

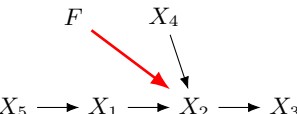

Figure 5: An example to show how Lemma 4.1 and 4.2 helps identify the root cause with a few invariance tests given a causal graph, where $X_1$ is the root cause.

## H  DISCUSSION ON THE TRADE-OFF BETWEEN SAMPLE COMPLEXITY OF LEARNING $\mathcal{C}$-ESSENTIAL GRAPH AND COMPUTATIONAL EFFICIENCY OF COMPUTING CMI

We will use Table 4 to illustrate the trade-off between the sample complexity and computational efficiency of the proposed algorithm RCG for RCA. We use $D$ to denote the ground truth DAG. We use $\mathcal{C}0$ to denote $\mathcal{C} = \{\{\emptyset\}\}$. We use $\mathcal{C}1$ to denote $\mathcal{C} = \{\{\emptyset\}, \{W\}, \{J\}, \{T\}, \{Z\}, \{Y\}, \{Q\}\}$. We use $\mathcal{C}2$ to denote $\mathcal{C} = \{\{\emptyset\}, \{W\}, \{J\}, \{T\}, \{Z\}, \{Y\}, \{Q\}, \{W, Z\}, \{W, Q\}, \{W, Y\}, \{W, J\}, \{W, T\}, \{Z, Y\}, \{Z, J\}, \{Z, T\}, \{Z, Q\}, \{Y, J\}, \{Y, T\}, \{Y, Q\}, \{J, T\}, \{J, Q\}, \{T, Q\}\}$. These sets are defined for Algorithm 3 to obtain the respective $\mathcal{C}$ essential graphs during the normal operation time and RCG can then take these graph objects as input for RCA post-failure.

We see that as we increase the number of conditioning sets in $\mathcal{C}$, the resulting $\mathcal{C}$-essential graph will become sparser. During the failure time, RCG will conduct an additional $n$ marginal invariance tests to further refine the graph objects shown by Table 4 depending on the defined $\mathcal{C}$. Thus, the possible parents of each observed variable will potentially get smaller. This will increase the computational efficiency and reduce the sample complexity of computing conditional mutual information in RCG during the failure time. However, as $\mathcal{C}$ gets larger, the sample complexity and time complexity also increase for using $\mathcal{C}$-PC during the normal operation time. Hence, there is a trade-off between learning $\mathcal{C}$-essential graphs during normal operation time and computing conditional mutual information post-failure in terms of sample and time complexity.

| Ground truth $D$ | Choice of $\mathcal{C}$ | $\mathcal{C}0$ | $\mathcal{C}1$ | $\mathcal{C}2$ |
|---|---|---|---|---|
| $W \quad J \rightarrow T$ 
 $\downarrow \quad \downarrow \quad \downarrow$ 
 $Z \quad Y \rightarrow Q$ | $\mathcal{C}$-essential graph | (graph) | (graph) | $W \quad J - T$ 
 $\downarrow \quad \downarrow \quad \downarrow$ 
 $Z \quad Y \rightarrow Q$ |

Table 4: A table that shows the trade-off between sample complexity and computational efficiency before and after failure for RCA of the proposed algorithm RCG using different $\mathcal{C}$ to learn $\mathcal{C}$-essential graphs.

## I  DISCUSSION ON CHALLENGES OF INCORPORATING $\mathcal{C}$-ESSENTIAL GRAPHS FOR RCA WITH CI TESTS ONLY

In this section, we first show how a partial causal graph represented by $\mathcal{C}$-essential graph learned from observed data before the failure period can facilitate an efficient RCA method with CI tests under the faithfulness assumption. Then, we discuss three difficulties of incorporating a $\mathcal{C}$-essential graph for RCA.

Given a $\mathcal{C}$-essential graph of a DAG $D_1$ shown in Figures 6(a) and 6(b) and by assumption A.16, we will show that it is possible to run a single CI test to identify the root cause during the fault period. To illustrate this concept, suppose an algorithm can pick on $X_1$ and test the CI relation $(F \perp\!\!\!\perp X_1)_P$. Since $X_2, X_3, X_4$ are non-ancestors of $X_1$ in $\varepsilon_{\{\{\emptyset\}\}}(D_1)$ and $(F \not\perp\!\!\!\perp X_1)_P$, one can infer that $X_1$ must be a child of $F$ in the ground truth. Hence, $X_1$ is the root cause.

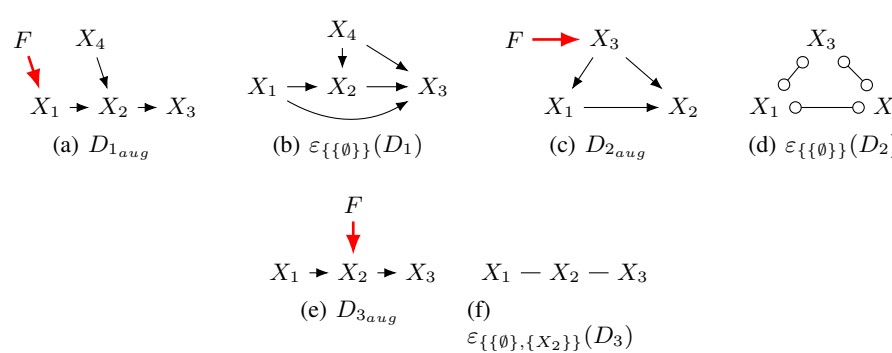

(a) $D_{1_{aug}}$     (b) $\varepsilon_{\{\{\emptyset\}\}}(D_1)$     (c) $D_{2_{aug}}$     (d) $\varepsilon_{\{\{\emptyset\}\}}(D_2)$

(e) $D_{3_{aug}}$     (f) $\varepsilon_{\{\{\emptyset\},\{X_2\}\}}(D_3)$

Figure 6: 6(a) - 6(b): an example shows how a $\mathcal{C}$-essential graph learned from observed data can be used to find root cause more efficiently where $\mathcal{C} = \{\{\emptyset\}\}$. 6(e) - 6(d): an example shows how a $\mathcal{C}$-essential graph may not help identify root causes with more CI tests since it does not have any orientations. 6(c) - 6(f): an example shows that not all $\mathcal{C}$-essential graphs that have no orientations are equally informative for RCA, where

In contrast, we will show how RCD (Ikram et al., 2022) is inefficient in terms of the number of CI tests used to identify root causes in this example and how the worst case for an algorithm that leverages partial causal structure still outperforms RCD in its best case. Suppose the ground truth DAG augmented by F-NODE is shown in Figure 6(a). Note that the best case for RCD must have tested 6 CI statements since the following CI statements must be observed based on the design of RCD in order to conclude $X_1$ to be root cause: $(F \perp\!\!\!\perp X_4)_P, (F \not\perp\!\!\!\perp X_2)_P, (F \perp\!\!\!\perp X_2|X_1)_P,$ $(F \not\perp\!\!\!\perp X_3)_P$ and $(F \perp\!\!\!\perp X_3|X_2)_P$ (or $(F \perp\!\!\!\perp X_3|X_1)_P$). Otherwise one will need to test CI relation between $F$ and $X_1$ by conditioning on all subsets of the power set of $\{X_2, X_3, X_4\}$. However, if we compare with the best case of an algorithm that leverages partial causal structure, it only requires to observe a single CI statement: $(F \not\perp\!\!\!\perp X_1)_P$. Note that even in the worst case, it only takes at most 4 CI statement, i.e., $(F \perp\!\!\!\perp X_4)_P, (F \not\perp\!\!\!\perp X_3)_P, (F \not\perp\!\!\!\perp X_2)_P, (F \not\perp\!\!\!\perp X_1)_P$ in order to conclude $X_1$ as it first searches through all marginal tests and can leverage the graph structure of $\varepsilon_{\{\{\emptyset\}\}}(D_1)$ learned from observed data.

However, there are a few challenges in incorporating a $\mathcal{C}$-essential graph for RCA. First, it is not clear how one should select a variable initially in a graph for testing conditional independence. Consider the same example in Figure 6(b), if $X_3$ is selected first instead of $X_1$ for testing the CI relation $(F \perp\!\!\!\perp X_3)_P$, then one should observe $(F \not\perp\!\!\!\perp X_3)_P$, implied by assumption A.16. Unfortunately, this test result does not eliminate the possibility that $X_3$ can be the root cause. It also does not give information to exclude $X_1, X_2, X_4$ from being the root cause. This shows that, given a $\mathcal{C}$-essential graph, the number of CI tests needed for RCA depends on both the graphical structure and the actual root cause location.

Second, some $\mathcal{C}$-essential graphs may not show any orientations. This posits a challenge that one may not hope to use fewer CI tests for RCA even when a partial causal structure is learned from observational data. For example, in Figure 6(d), $F$ is d-connecting with all observed variables. Unlike the example in Figure 6(b), even when we have exhausted all marginal CI tests among the observed variables and $F$ during the failure period, we cannot utilize any ancestral relationships in the graph structure to determine which variable cannot be the root cause.

Third, all $\mathcal{C}$-essential graphs that do not have any orientations may not be equally informative for RCA. For instance, if the $\mathcal{C}$-essential graph is the graph shown in Figure 6(f), according to Figure 6(e), we see that $(F \perp\!\!\!\perp X_1)_P$ and $(F \not\perp\!\!\!\perp X_2)_P$ hold based on assumption A.16. One can infer that i.) $F$ cannot point to $X_1$ due to $(F \perp\!\!\!\perp X_1)_P$, ii.) $F$ does not have a directed path to $X_1$ and iii.) $F$ has a directed path to $X_2$. Therefore, $X_1 - X_2$ can further be oriented as $X_1 \to X_2$ in Figure 6(f) with interventional data. Since all the unshielded colliders in Figure 6(f) should have been oriented by $\mathcal{C}$-PC (see line 4 in Algorithm 3), $X_2 - X_3$ can then be further oriented as $X_2 \to X_3$, resulting in $X_1 \to X_2 \to X_3$. Hence, we can conclude $X_2$ to be the root cause as $X_2$ is the parent of $X_3$. As such, the $\mathcal{C}$-essential graph in Figure 6(f) is more informative than the one in Figure 6(d) for RCA.

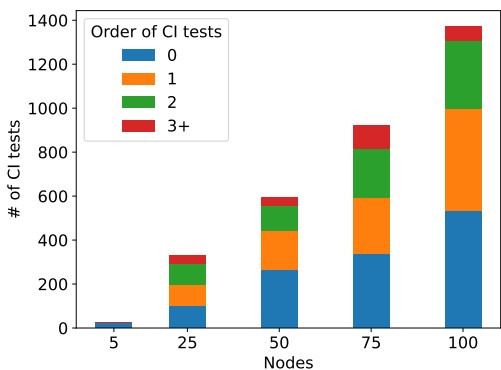

Figure 7: The number of CI tests executed by RCD and the size of the separating set used in those tests. As the number of nodes increases, RCD relies on higher-order CI tests to identify the root cause. However, these higher-order tests are less reliable with limited samples, which diminishes RCD's effectiveness.

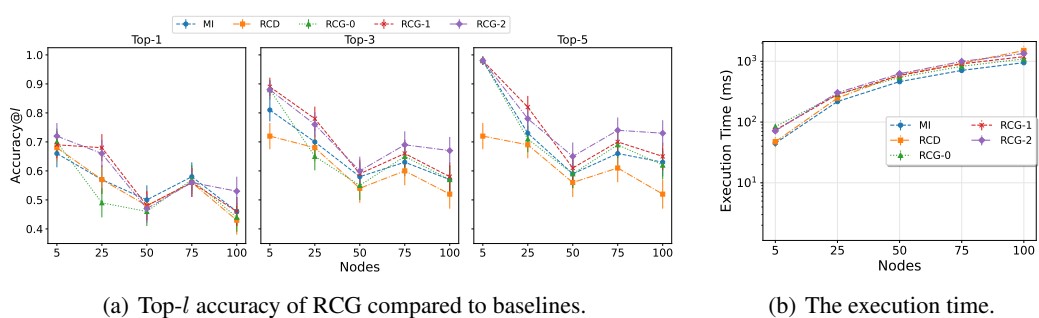

(a) Top-$l$ accuracy of RCG compared to baselines.  (b) The execution time.

Figure 8: Top-$l$ accuracy and the runtime of RCG compared to the baselines. The input graph in this experiment were learned from the data using $\mathcal{C}$-PC.

## J ADDITIONAL EXPERIMENTS

### J.1 RCD WITH HIGHER-ORDER CI TESTS

Figure 7 illustrates the number of CI tests executed by RCD alongside the size of the separating sets used. RCD identifies the root cause by gradually increasing the size of these sets. However, the statistical power of CI tests diminishes with larger separating sets, particularly when sample sizes are limited, as is often the case in RCA, where quick failure resolution is crucial (Shah & Peters, 2020; Kocaoglu, 2023). This reliance on higher-order CI tests leads to poorer performance with an increasing number of nodes, as discussed in Section 6 of the main paper. In contrast, RCG mitigates this issue by using $\mathcal{C}$-PC, which is more effective than full graph learning, and after a failure, it relies solely on $n$ marginal invariance tests.

### J.2 EXPERIMENTS WITH SAMPLED VERSION

Figure 8 illustrates the performance of RCG in comparison to MI and RCD. Similar to the experiment using the ground truth causal graph, we utilized 10,000 samples for the observational dataset and only 100 samples for the interventional dataset. Additionally, we included RCG-0 and RCG-1 to demonstrate the performance across different values of $k$ for $\mathcal{C}$-PC, where $k$ determines the size of the maximum separating set within $\mathcal{C}$. We did not include RUN in this experiment, as it requires continuous data, while our dataset in this experiment is discrete. Furthermore, RCG(IGS) and RCG(CPDAG) were omitted since we cannot derive a complete DAG from the samples, and learning the full CPDAG from the samples is exceedingly time-consuming Ikram et al. (2022).

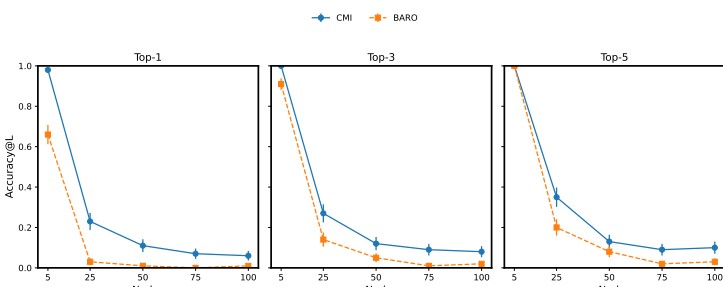

Figure 9: Average top-$l$ accuracy of CMI compared to BARO with $l \in 1, 3, 5$ over 100 repeated experiment per graph size. The results demonstrate that CMI with correct causal knowledge consistently provides better accuracy compared to the state-of-the-art algorithm BARO even when the distribution is only continuous. Both observational and interventional sample sizes are $100n$ for $n \in \{5, 25, 50, 75, 100\}$.

The results align with our earlier findings presented in the main paper. RCD exhibits poor performance because it lacks access to causal relationships, leading it to condition on all nodes until a separator is found. This results in lower accuracy for RCD. In contrast, RCG yields better results as the value of kk increases. Notably, RCG-1 and RCG-2 consistently outperform RCD, while RCG-0 occasionally produces results similar to RCD, but sometimes fails to identify the root cause. This inconsistency arises because RCG-0 struggles to learn a sufficiently sparse graph, resulting in conditioning on a larger set of nodes, which diminishes the reliability of the conditional independence test.

### J.3 Linear Gaussian Additive Models with BARO

We want to demonstrate the merit of correct causal knowledge and do so by comparing our proposed method with access to a correct directed acyclic graph of the underlying system with one of the state-of-the-art methods called BARO Pham et al. (2024).

As BARO is restricted to data where median and interquartile range can be computed, we provide a synthetic experiment that generates DAGs of size $n \in \{5, 25, 50, 75, 100\}$. The sample size for observational data and interventional is proportional to the number of variables e.g. $100n$. A root cause is randomly assigned and there are at least $k$ descendants randomly assigned to the root cause where $k > 0$. Then, there is a probability of $0.7$ that there exists a confounder between the root cause and one of its descendants. Then, directed edges are randomly assigned between a pair of nodes that are not the root cause and its descendants with a probability of $0.6$ while acyclicity is maintained. Each variable that has no parents follows a standard Gaussian distribution. Any variable that has parents will take a weighted sum of its parents with an additive standard Gaussian noise. The weight from each parent is sampled from a uniform distribution between $0.5$ and $1.5$ over the size of the graph. If the root cause variable does not have any parents, then it follows a Gaussian distribution with mean sampled from a uniform distribution between $-10$ and $10$ and standard deviation sampled from a uniform distribution between $1.5$ and $10$. Otherwise, it is a weighted sum of its parents plus a noise term that follows a Gaussian distribution with mean sampled from a uniform distribution between $-10$ and $10$ and standard deviation sampled from a uniform distribution between $1.5$ and $10$. We repeat the experiment for 100 times per graph size. We provide the exact index of the data point that follows the interventional distribution for BARO. We discretize the dataset with $k$-bins discretizer in `scikit-learn` (Pedregosa et al., 2011) with the setting: $k = 3$, `encode = ordinal`, `strategy=kmeans`. We compute $I(X; F | PossPa_D(X))$ by counting the frequencies for each node $X$ given a correct DAG. We rank each node by sorting $I(X; F | PossPa_D(X))$ for each $X$ in descending order. This approached is denoted as CMI. We limit both the observational and interventional sample sizes to 100 for each size of the graph.

From Figure 9, we see that the use of conditional mutual information with the correct causal knowledge consistently outperforms BARO under a limited sample across all graph sizes. There is almost $0.4$ average top-1 accuracy difference for the small graph of size 5. The difference becomes small

as the graph size increases. This is expected as the data gets noisier with larger graphs due to the experimental setup. We see that our approach achieves $100\%$ average top-3 accuracy for graphs with 5 variables. This experiment shows the benefits of having correct causal knowledge in the presence of spurious correlation.

