# OpenReview forum: "Root Cause Analysis of Failure with Observational Causal Discovery"
_ICLR.cc/2025/Conference — Submitted to ICLR 2025_

### Official Review · Reviewer_6ic8 · 2024-10-22

**Soundness:** 2
**Presentation:** 2
**Contribution:** 2
**Rating:** 5
**Confidence:** 3

**Summary:**

This research links root cause analysis in network failures to Interactive Graph Search (IGS), establishing a mathematical lower bound for the number of tests needed. With a fully known causal graph, the authors then propose an optimal algorithm that achieves this bound. With a partially known causal graph, they can identify the root-cause with linear number of invariance tests.

**Strengths:**

1. The motivation of this work makes sense. I totally agree that RCA is time-sensitive only after the failure occurs. We can use the time before a failure to learn the causal graph which can help us to reduce the number of tests of conditional independence in the period of RCA.

2. The authors provide extensive experimental results and detailed discussion. In particular, I very appreciate Appendix I.

**Weaknesses:**

1. This work relies heavily on previous works. First and foremost, it borrows the idea of modeling a failure as an intervention and transforming RCA into a problem of finding adjacency of the F-NODE from (Ikram et al., 2022). Also, it directly uses the theoretical results in (Shangqi et al., 2023) and C-PC in (Lee et al., 2024). More specifically, (Ikram et al., 2022) has already linked RCA to causal discovery and most causal discovery techniques used in this paper are also proposed by previous works. In my opinion, the major contribution in causal discovery of this paper lies in Lemma 4.1, 4.2, and 5.2. Considering that the authors list "causal discovery" as the first keyword, I think their contribution in this aspect is limited.

2. The organization of this paper should be improved. The discussion on related works is spread across many sections. The authors can use a dedicated section to introduce existing techniques used in this paper and the detailed differences between this work and previous works, rather than giving too many details in Sec. 1, 4, 5, which makes it harder for readers to grasp their contributions. Besides, I strongly suggest the authors move Appendix I to the main text.

3. Some minor concerns are detailed in Questions.

If the authors can address my concerns, I would like to raise my score.

**Questions:**

1. The authors claim that RCD performs an exponentially large number of CI tests, but I'm not sure this is correct for Hierarchical Learning in (Ikram et al., 2022).

2. Lemma 4.1 and 4.2 are both based on the fact there is only one single root cause, but it is possible that there are multiple root causes in real-world scenarios, limiting applicability of this work.

3. There is too much space between references in page 14.

---

> ### Author Response · Authors · 2024-11-21
>
> We sincerely appreciate the reviewer’s feedback and thoughtful questions. We will try to address reviewer's key concerns:
>
> > This work relies heavily on previous works...
>
> We agree with the reviewer that our solution leverages several existing methods and findings. We also acknowledge that our main technical contributions lie in Lemmas 4.1, 4.2, and 5.2, with **Lemma 5.3 being the most important**. We apologize for mistakenly using "causal discovery" as the keyword, as the reviewer correctly points out that our primary contribution is in identifying novel ways of finding the root cause given a partial causal structure. For convenience, we are providing a short summary of our contributions in this paper:
>
> - We provided a lower bound for any algorithm that uses only marginal independence tests for finding a single root cause by reducing the problem of RCA to interactive graph search (IGS) given a causal DAG.
>
> - Under Lemma 5.3, every variable that is not a root cause must be d-separated with F-nodes given their possible parents in the fine-grained C-essential graph after incorporating marginal invariance tests. We connect this insight with conditional mutual information (see Proposition 5.1) to easily rank the root causes. This contrasts with many existing works. For example, RUN or CausalRCA impose a weight on each edge via heuristics and use PageRank to sort for top $l$ root causes. Similarly, RCD arbitrarily increases the threshold of alphas for CI tests to obtain the top-$l$ root causes, but the notion of ranking among the reported variables is unclear, as they rely on p-values from statistical tests.
>
> - Our work leverages a partial causal graph learned from observed data to facilitate root cause analysis. This is non-trivial in terms of reducing the number of CI tests (see Appendix I) as there is a trade-off between exploring useful orientations and testing for adjacency between F-nodes and observed variables, as seen in RCD. In contrast, we only need n marginal invariance tests to exploit such a partial causal structure. To the best of our knowledge, prior to our work, it was not known whether incorporating a partial causal structure would be beneficial for RCA.
>
> Finally, we believe our empirical observations are both surprising and instructive. Specifically, we observe that sequential root-cause discovery (RCG(IGS)), which theoretically benefits from a logarithmic number of tests, struggles in practice. In contrast, using a linear number of tests, when paired with a partial causal graph, yields much better practical performance. We believe this is an important insight, which might be known within the causal discovery community, but not known in RCA literature.

---

> > ### Comment · Reviewer_6ic8 · 2024-11-27
> >
> > Thanks for your response. My expertise lies primarily in causal discovery, and I maintain my earlier assessment that the contributions of this work to the community of causal discovery are relatively limited. It does not provide new identifiability results or more advanced causal discovery method. However, the authors argue that their work makes novel and significant contributions to RCA, an area where I lack sufficient expertise to make a reliable evaluation. I'd like to raise my score to 5 and decrease my confidence to 3.

---

> > > ### Author Response · Authors · 2024-11-27
> > >
> > > We sincerely thank the reviewer for raising the score and for providing constructive feedback to help us improve the paper.

---

> ### Author Response · Authors · 2024-11-21
>
> > The organization of this paper should be improved...
>
> We sincerely thank the reviewer for the suggestions. We will polish the draft accordingly.
>
> > The authors claim that RCD performs an exponentially large number of CI tests, but I'm not sure this is correct for Hierarchical Learning in (Ikram et al., 2022).
>
> We apologize for the confusion. We meant that it’s exponential in terms of the size of the partitions for RCD.
>
> > Lemma 4.1 and 4.2 are both based on the fact there is only one single root cause...
>
> For concerns regarding the assumption of a single root cause, please refer to our general comment. Furthermore, it can be observed that even with real-world application, our proposed RCG outperforms a recently developed state-of-the-art algorithm known as BARO. Additionally, RCG’s ability to rank the nodes allows it to identify multiple root causes, demonstrating its flexibility in handling cases with more than one root cause.
>
> > There is too much space between references in page 14.
>
> Thanks for the suggestion. We will fix it.

---

### Official Review · Reviewer_Aawg · 2024-10-29

**Soundness:** 3
**Presentation:** 2
**Contribution:** 2
**Rating:** 5
**Confidence:** 4

**Summary:**

The paper proposes an approach for identifying the root cause in a system where we observe anomalies. For this, the authors propose to utilize data from the normal operating regime to infer a (partial) causal graph. The graphical information is then used to reduce the number of independence tests to identify the root cause node based on the assumption of a shift in the underlying data generating mechanism. The approach has been evaluated using artificial and real-world data.

**Strengths:**

+ Insightful analysis and fair discussion of causal discovery in a large-scale setting
+ Good introduction to the problem
+ Extensive additional information in appendix for certain details

**Weaknesses:**

- While the related work section has a fair discussion about different works, it also lacks work involving the direct use of graph structure (see Questions section for more details).
- The proposed method certainly has its novelty, but it seems rather limited as it boils down to the idea of first reconstructing the causal graph using causal discovery and by this, naturally, reducing the search space when running independence tests. The arguments for papers that address the problem without graph knowledge explicitly avoid a causal discovery step.
- Some definitions (like SCMs) are introduced but then not really needed. A shift in the mechanisms can be defined without this.
- The formulation of some of the definitions could be improved (e.g., when introducing a causal graph). However, these are minor issues that could be easily fixed in a revision.
- Some assumptions are not clearly stated and implied. For instance, the assumption that there can only be one root cause.

For more details, see the Questions section.

**Questions:**

The work certainly has some novel and great insights. However, I am concerned about the (more high-level) novelty here. The related work that identifies the node with a mechanism shift without assuming a graph naturally needs to run this on a potentially exponential number of combinations. Their main claim is also about not needing the graph structure, as this is an obvious way to reduce the required number of independence tests one needs to perform. Running causal discovery on the normal operation period of a system is a logical first step if a method requires a causal graph or, as in your case, to reduce the search space. In that sense, I am concerned about the novelty claim that one needs fewer tests if the graph is given, as this is obvious. I might be missing a crucial part here in the, admittedly, over-simplification of the idea and hope the authors can comment on this.

Some further remarks:
- The related work focuses on certain types of work in the domain of root cause analysis but lacks discussion about other types of work that utilize a causal graph directly, such as:

"Identifying patient-specific root causes with the heteroscedastic noise model" by Strobl et al.
"Causal structure-based root cause analysis of outliers" by Budhathoki et al.
"Counterfactual formulation of patient-specific root causes of disease" by Strobl et al.
"Root Cause Analysis of Outliers with Missing Structural Knowledge" by Okati et al.

- The difference between the complexities mentioned on lines 101 and 103 is not clear, and further clarification would be helpful.
- In Definition 2.1, the formal definition of the graph is lacking, which you then later use in Assumption 2.3. You could move this to Definition 2.1 already.
- The notation Z(X, Y ∉ Z) in line 130 is confusing; can you clarify this?
- As mentioned before, the need to introduce SCMs is unclear as a mechanism shift can also be purely introduced using the Bayesian network formulation.
- A clear assumption statement that you assume a single root cause is lacking.
- The faithfulness assumption is important for causal discovery via CIs, but that connection could be emphasized more clearly.
- Applying causal discovery on the 'normal operation' period alone implies the assumption that the causal structure has changed in the anomalous regime. While this is a valid assumption, it is also only made implicitly. In a general setting, anomalous data can even be particularly helpful in identifying cause-effect relationships.
- The introduction of the notation for having multiple metrics for a node over time does not seem to be used afterward. While I am not very familiar with the C-PC algorithm, it is unclear how one would perform causal discovery in such a setting with high-dimensional nodes and temporal dependencies without employing more time-based causal discovery approaches. Does the data you used in the experiments reflect such data?
- A drawback of your approach that lacks further discussion is the requirement of a "sufficiently" large sample size of the anomalous population. Since you argue that the root cause needs to be identified in a timely manner, this would only work in a system that produces a lot of data. If, for example, the system only produces a metric observation every few minutes, you would not have enough samples. This aspect could be discussed further as the works mentioned in the first point would work on single observations.

---

> ### Author Response · Authors · 2024-11-21
>
> We sincerely thank the reviewers for their detailed and insightful feedback. Below, we address the key points raised:
>
> > While the related work section has a fair discussion about different works, it also lacks work involving the direct use of graph structure ...
>
> We thank the reviewer for the suggestions. We will include more discussion on the related work given. Here, we briefly discuss the difference between the papers mentioned by the reviewer and our work. Many of these methods impose assumptions that are not aligned with our problem setup:
>
> - **Budhathoki et al.** assume a DAG with all functional relationships explicitly provided.
> - **Strobl et al.** rely on additive noise models with restrictive assumptions, such as non-Gaussian error terms or invertible SCMs.
> - **Okati et al.** propose a series of methods in settings ranging from known to unknown causal structures. They also assume an additive noise model. Also, their approach assumes there is only one single data point in the anomalous regimes. It is not clear how the approach handles more than a single data point in general and it does not necessarily suit our problem setup. We will demonstrate how a method that relies on a single data point can be just as good as a method that randomly chooses the root cause. We will share the results here once we have them.
>
> > I am concerned about the novelty claim that one needs fewer tests if the graph is given, as this is obvious...
>
> While it might seem intuitive that causal graphs can help reduce the search space, incorporating causal knowledge for RCA introduces non-trivial challenges, particularly due to the uncertainty of the interventional target. For example, when working with interventional data, we face the decision of whether to explore more informative orientations in a partially oriented causal graph or, as with RCD, to exploit the adjacency between the F-node and observed variables through conditional independence (CI) tests. Our algorithm addresses this dilemma by systematically exploring the graph. A more detailed discussion of this approach is provided in Appendix I.
>
> > The difference between the complexities mentioned on lines 101 and 103 is not clear...
>
> Here, we provide a lower bound for the number of marginal invariance tests required by any algorithm that relies solely on marginal invariance tests to identify root causes. This is achieved through a reduction from RCA to another problem known as interactive graph search (IGS).
>
> > The notation $Z(X, Y \not \in Z)$ in line 130 is confusing; can you clarify this?
>
> We apologize for the confusion. This is simply a reminder that $Z$ does not contain the pair of variables $X$ and $Y$ in the definition.

---

> ### Author Response · Authors · 2024-11-21
>
> > As mentioned before, the need to introduce SCMs is unclear...
>
> SCMs and Bayesian Networks are both graphical models used for probabilistic reasoning, but they serve different purposes and have distinct characteristics, particularly in relation to causality. SCMs explicitly model causal relationships, which enables causal inference. They consist of structural equations that define how each variable is determined by its parents, making it possible to analyze interventions. In contrast, Bayesian networks represent conditional dependencies among variables without necessarily implying causation. The concept of interventions is central to our paper, as we model system failures as interventions to the root cause. Therefore, we include SCMs in the main paper. We will clarify this distinction further in the revised manuscript.
>
> > A clear assumption statement that you assume a single root cause is lacking.
>
> We apologize for the confusion. The single root cause assumption is only used in the IGS method, specifically for Lemma 4.1 and Lemma 4.2. Our method for the unknown graph case _does not_ rely on this assumption. However, we will clarify this distinction in the manuscript. For further details about single root cause assumption, please refer to our general comment.
>
> > The faithfulness assumption is important for causal discovery via CIs, but that connection could be emphasized more clearly.
>
> We will make this more clear. Thanks for the suggestion.
>
> > Applying causal discovery on the 'normal operation' period alone implies...
>
> We apologize for the confusion and will briefly explain the problem setting. The causal structure is a DAG augmented with an F-node, where we assume that F points to the root causes in the underlying data-generating process. The causal structure itself does not change from normal operation to the anomalous regime. The only difference is that F takes the value 0 during normal operation and 1 during the anomalous regime. For a formal treatment, please refer to Section 4 of Ikram et al. (2022).

---

> ### Author Response · Authors · 2024-11-21
>
> > The introduction of the notation for having multiple metrics for a node over time does not seem to be used afterward...
>
> The number of variables corresponds to the number of metrics involved in RCA. We currently assume the data to be iid and first discretize the data before applying $\mathcal{C}$-PC. However, this method is not suitable for high-dimensional variables, such as images. That said, our pipeline is not restricted to $\mathcal{C}$-PC; it can accommodate any algorithm that outputs an essential graph. Additionally, the real-world data we used exhibits time dependence. We demonstrate the utility of our method through experiments on Sock Shop data and real-world application data, showing that our method outperforms state-of-the-art techniques in these real-world scenarios.
>
> > A drawback of your approach that lacks further discussion is the requirement of a "sufficiently" large sample size of the anomalous population...
>
> We appreciate the emphasis on sample complexity and will address it more explicitly in the manuscript. Our algorithm leverages marginal independence tests during post-failure time, which are more sample-efficient than the conditional independence tests used in our baseline (RCD). This design choice ensures robustness even with limited samples. We include a discussion on the trade-offs between sample complexity and computational efficiency in Appendix H. By using lower-order CI tests, as recommended in prior work (e.g., [1]), RCG strikes a balance between reliability and computational cost. We also provide runtime comparisons with RCD in our experiments, demonstrating comparable performance while maintaining sample efficiency.
>
> [1] Kocaoglu, Murat. "Characterization and learning of causal graphs with small conditioning sets." _Advances in Neural Information Processing Systems_ 36 (2024).

---

> > ### Author Response · Authors · 2024-11-24
> > **Further comparison with the related work mentioned**
> >
> > - We sincerely thank the reviewer for highlighting a relevant field of study. Following your suggestion, we chose [1] to evaluate the applicability of our solution. We were particularly intrigued by the baseline that requires only a single failure sample to identify the root cause. To this end, we implemented their proposed method, **SCORE ORDERING**, which uniquely does not rely on a causal graph or SCM and instead uses estimated anomaly scores.
> >
> > - We would like to point out that, without further assumptions on the data-generating process, in the fully non-parametric regime that we operate in, we do not expect the method to significantly outperform random guess. We put this hypothesis to the test and report our results below. Specifically, we compare **SCORE ORDERING** with the random scheme and observe that **SCORE ORDERING** performs only marginally better than random selection. Furthermore, their method assumes an invertible SCM with additive noise, while our method does not rely on any such assumptions. Their limited ability to identify root causes in real-world scenarios is evident even in their own results (Table 3 in [1]), where the top-1 recall is reported as $0.1$.
> >
> > | Nodes | SCORE ORDERING  | Random |
> > |-------|-------|--------|
> > | 5     | 0.3   | 0.24   |
> > | 10    | 0.15  | 0.06   |
> > | 15    | 0.17  | 0.1    |
> > | 20    | 0.08  | 0.03   |
> > | 25    | 0.08  | 0.05   |
> >
> > - In contrast, we demonstrate the robustness of RCG not only with synthetic datasets but also with a real-world application and a production-level dataset, showcasing its broader applicability.
> >
> > - We hope these findings address the reviewer's questions and assist in supporting the acceptance of our paper.
> >
> >
> > Reference:
> >
> > [1] Okati, Nastaran, Sergio Hernan Garrido Mejia, William Roy Orchard, Patrick Blöbaum, and Dominik Janzing. "Root Cause Analysis of Outliers with Missing Structural Knowledge." arXiv preprint arXiv:2406.05014 (2024).

---

> > > ### Comment · Reviewer_Aawg · 2024-11-26
> > >
> > > I want to thank the authors for their response, as some of my concerns were addressed. I certainly appreciate the practical value of the provided algorithm, but the theoretical novelty over existing work still appears rather limited. I understand the argument regarding unknown target variables, but RCA approaches based on causal graphs (even if we assume causal discovery is part of the RCA process) have the same benefit of reduced search spaces. The paper could benefit from more clearly pointing out the theoretical advantages (besides the empirical evaluation) over causal graph-based RCA approaches in settings where, e.g., the causal graph is known beforehand.

---

> ### Author Response · Authors · 2024-11-27
>
> We sincerely thank the reviewer for the thoughtful engagement. Could the reviewer kindly elaborate on the following statement?
> > "RCA approaches based on causal graphs (even if we assume causal discovery is part of the RCA process) have the same benefit of reduced search spaces"?
>
> To the best of our knowledge, no existing RCA method incorporates a partial causal structure. Moreover, it is not clear how many CI tests are required to identify a root cause using an RCA method based solely on CI tests. We would greatly appreciate any further insights or references the reviewer could provide.
>
> Once again, we thank the reviewer for spending time with us.

---

> > ### Comment · Reviewer_Aawg · 2024-11-27
> >
> > > To the best of our knowledge, no existing RCA method incorporates a partial causal structure.
> >
> > This is a fair point as I was more considering non-CI based approaches that could provide the full DAG, i.e., any graph-based approaches would benefit from this structural knowledge. However, I agree that the point of using partial knowledge with respect to a CI-based approach is a valuable contribution. Therefore, I am willing to increase my score.

---

> > > ### Author Response · Authors · 2024-11-27
> > >
> > > Thank you for revisiting and updating your review score for our paper. We sincerely appreciate the time and effort you’ve dedicated to assessing our work. Your feedback has been invaluable in helping us refine our research and presentation. If there are any further clarifications or questions we can address, please don't hesitate to let us know.

---

### Official Review · Reviewer_93kt · 2024-10-30

**Soundness:** 2
**Presentation:** 2
**Contribution:** 2
**Rating:** 3
**Confidence:** 4

**Summary:**

Using causal analysis, the authors provide and review methods to determine the root cause(s) of failure for up to 100 nodes.

**Strengths:**

An attempt is made here to give an algorithm for root cause analysis that considers some of the literature. Preliminary results are promising.

**Weaknesses:**

This paper needs more work. There are claims throughout that seem like they're not quite thought through. I will list a few, but generally, the paper needs to focus more on comparing the best available methods in the literature and be rewritten with this in mind. That requires a bit more work than was taken up here.

For example, the reference to Chickering 2004 as evidence that causal search algorithms are slow is a bit forced since there are implementations of Chickering 2002 that are quite fast (e.g., FGES). The Lam et al. 2022 paper referenced is pretty slow for 100 nodes, but a follow-up paper to this in Neurips 2023 is quite fast for 100 nodes and scales to 1000 nodes with near-perfect accuracy. Also, whether a causal search algorithm is slow largely depends on the model class. For the linear Gaussian or multinomial cases, algorithms can be quite fast, but general distributions can become very slow, as general tests or scores need to be employed. The speed and accuracy also depend on the density of the graph. FGES (above) is very accurate for sparse graphs (sparsity = 1 - avg degree / # nodes, so for 100 nodes, average degree 4 might be considered sparse). But for dense graphs, its accuracy falls off quickly. PC (and derivatives) tend to have decent adjacency precision, but adjacency recall can be low, and orientation accuracy can be low. The devil is in the details. So those comments were a little too hand-wavy. For the version of PC you're using, you need to cite accuracy statistics not just for adjacency but also for orientation, as you are making claims about whether ancestral paths exist. This is completely missing from the draft.

As a general strategy, one should compare one's novel methods to _the best-performing alternative methods in the literature_, not just a few chosen methods one has on hand. As for the methods compared, these don't seem like the best methods that could be devised in the literature, so more work needs to be done to find what those methods might be (or devise them) and compare them. The PC version you're using should be compared to other plausible alternatives, such as the one mentioned above, or to the R method, BIDAG, which is also quite good. Again, for timing results, just give the _actual timing results_ for the various methods and let the reader decide. If the C-PC method turns out to be the winner, this should be evident from one's figures.

In addition, there are more papers on root cause analysis than are given in the literature review; this could well be expanded.

Some minor comments.

1. The definition of Markov given is for DAGs in particular, not for arbitrary graphs. It doesn't even work for what you're calling "essential graphs."

2. There is a little confusion about soft intervention. If you do a "soft intervention" on a variable X, X can still have parents. The case where it cannot have parents is where you have a "hard intervention," in which case you replace its distribution with a parent-free distribution of your choice. This is a terminological problem that can be fixed.

There is a little confusion between the lemmas given on p. 4 and the algorithms later in the paper. On p. 4, you claim that "The following two lemmas use the fact that there is only one single root cause," leading me to think that you are only considering the case where there is a single root cause of failure in the system. It's a strong assumption, but fair enough. But later, in the algorithms, you say you list the top l root causes. I could not discern any transition between these two ideas.

Typos p. 5 "Algorithm the only" ==> "Algorithm that only"; "C avid" ==> "C to an avid"

You say proofs are to be left to the avid reader, but for a paper for ICML, you should supply the proofs of your claims.

In Algorithm 2, circle endpoints suddenly appear out of nowhere. What are these? Are you dealing with PAGs here?

**Questions:**

Would you be able to try various other methods besides C-PC for the essential graph search, trying to find the best performers in the literature?

Would you be able to go through the literature more thoroughly and give a more substantive literature review?

---

> ### Author Response · Authors · 2024-11-21
>
> We thank the reviewer for the suggestions. We will address each concern below:
>
> > ...the reference to Chickering 2004 as evidence that causal search algorithms are slow is a bit forced since there are implementations of Chickering 2002 that are quite fast (e.g., FGES)...
>
> The reason we use $\mathcal{C}$-PC in our proposed algorithm is to highlight that our algorithm can accept a wide range of Markov equivalence classes of DAGs characterized by $\mathcal{C}$-essential graphs. It is important to note that the notion of essential graphs is subsumed by $\mathcal{C}$-essential graphs when $\mathcal{C}$ is defined to include all conditioning sets.
>
> The choice of discovery algorithm depends on various factors, such as sample size, the number of variables, and the underlying model assumptions. Our algorithm is not limited to only the CPC method, and selecting the best algorithm in general can be challenging. To address this, in Figure 2a, we demonstrate the potential performance if a perfectly accurate essential graph is obtained, as shown by the algorithm labeled RCG (CPDAG). This provides an indication of the expected performance when using the best causal discovery algorithm that generates an essential graph.
>
> > there are more papers on root cause analysis than are given in the literature review; this could well be expanded.
>
> Thank you for your suggestions. We will add more discussion to the related work in root cause analysis.
>
> > The definition of Markov given is for DAGs in particular, not for arbitrary graphs...
>
> We are not sure which part of the paper the reviewer is referring to; could you please clarify? We assume the ground truth is a DAG, and the conditional independence relations are induced from the DAG via the causal Markov condition. To learn the $\mathcal{C}$-essential graph, we rely on the faithfulness assumption.
>
> > There is a little confusion about soft intervention...
>
> We apologize for any confusion caused. In the manuscript, we clarify that incoming edges are removed for hard interventions, while the original causal graph is retained in the case of soft interventions (see Section 2, Line 136).

---

> ### Author Response · Authors · 2024-11-21
>
> > ...leading me to think that you are only considering the case where there is a single root cause of failure in the system. It's a strong assumption, but fair enough. But later, in the algorithms, you say you list the top l root causes…
>
> We apologize for the confusion. Indeed, the result for the algorithm named IGS in the known graph section relies on the assumption of a single root cause. However, our proposed algorithm, RCG, which takes a C-essential graph as input, is not restricted to the case of having a single root cause when a DAG is not provided. For more details about single root cause assumption, please see our general comment.
>
> Regardless of the actual number of root causes in the DAG, the RCA literature employs a metric called top-$l$ accuracy, where the output is a list of $l$ potential root causes, even if there is only a single root cause. Therefore, when comparing RCG to other approaches, we output $l$ nodes as potential root causes. We again apologize for the confusion and will update the manuscript to reflect this difference.
>
> > You say proofs are to be left to the avid reader, but for a paper for ICML, you should supply the proofs of your claims.
>
> Thank you for the suggestion. We put most of the proofs in the appendix due to space limits.
>
> > In Algorithm 2, circle endpoints suddenly appear out of nowhere. What are these? Are you dealing with PAGs here
>
> We apologize for the confusion. We explain the notations (see Definition A.13) and concepts of C-essential graphs (See Definition A.14) in the appendix.
>
> > Would you be able to try various other methods besides C-PC for the essential graph search ...
>
> Yes, we will compare the methods you suggested to evaluate whether performance improves in the experiment with finite observational data. We will share the results here once we have them.
>
> We will fix the typos in the manuscript and thank the reviewer for their details comments.

---

> > ### Author Response · Authors · 2024-11-24
> > **Updates on the experiments**
> >
> > - We sincerely thank you for providing valuable baselines that have helped us explore essential graph learning for our problem setup. Based on your review, we believe you are referring to the NeurIPS 2023 paper on **BOSS** [1], a successor to **GRaSP** [2]. Using the **BOSS** implementation from the causal-learn library, we successfully reproduced the authors' results. Specifically, with continuous datasets, **BOSS** efficiently learned an essential graph of $1000$ nodes in under $20$ seconds.
> >
> > - However, in our setting, where 1) the data does not follow a specific distribution and 2) the data is discrete, **BOSS** faces challenges. For instance, when using our discrete, randomly generated dataset, we had to switch **BOSS**’s score function to 'local_score_BDeu' from 'local_score_BIC_from_cov', which is designed for discrete data. With this adjustment, **BOSS** required approximately three hours to learn the essential graph for just $50$ nodes. We hypothesize that this inefficiency stems from **BOSS**'s linear Gaussian assumptions.
> >
> > - That said, we believe **BOSS**, or any similar method for learning essential graphs, could be seamlessly integrated into our proposed method. We chose **$\mathcal{C}$-PC** [3] to demonstrate that our method can handle diverse equivalence classes characterized by conditional independence constraints. This is possible because the $\mathcal{C}$-essential graph generalizes both essential and $k$-essential graphs [4], enabling integration with methods like the one suggested by reviewer 93kt.
> >
> > - We hope these findings address the reviewer's questions and assist in supporting the acceptance of our paper.
> >
> > Reference:
> >
> > [1] Andrews, Bryan, Joseph Ramsey, Ruben Sanchez Romero, Jazmin Camchong, and Erich Kummerfeld. "Fast scalable and accurate discovery of dags using the best order score search and grow shrink trees." Advances in Neural Information Processing Systems 36 (2023): 63945-63956.
> >
> > [2] Lam, Wai-Yin, Bryan Andrews, and Joseph Ramsey. "Greedy relaxations of the sparsest permutation algorithm." In Uncertainty in Artificial Intelligence, pp. 1052-1062. PMLR, 2022.
> >
> > [3] Lee, Kenneth, Bruno Ribeiro, and Murat Kocaoglu. "Constraint-based Causal Discovery from a Collection of Conditioning Sets." 9th Causal Inference Workshop at UAI 2024. 2024.
> >
> > [4] Kocaoglu, Murat. "Characterization and learning of causal graphs with small conditioning sets." Proceedings of the 37th International Conference on Neural Information Processing Systems. 2023.

---

> > > ### Author Response · Authors · 2024-11-30
> > >
> > > Dear Reviewer 93kt,
> > >
> > > As the discussion period draws to a close, we wanted to check if there are any concerns that we may not have addressed. We greatly value your time and effort in reviewing our paper and providing thoughtful feedback.

---

### Official Review · Reviewer_yWZE · 2024-10-31

**Soundness:** 2
**Presentation:** 2
**Contribution:** 2
**Rating:** 5
**Confidence:** 4

**Summary:**

The authors provide a method for identifying the root cause based on marginal invariance tests. The proposed method includes two steps: first, recovering the skeleton of the underlying causal model using normal (pre-failure) data, and second, constructing an augmented graph using invariance tests to identify the root cause by computing conditional mutual information. The authors demonstrate that, if the underlying causal model is known, the root cause can be identified using $O(\log(n))$ marginal invariance tests, where $n$ is the number of observed variables. Additionally, given observational data, the root cause can be recovered using $O(n)$ invariance tests according to the proposed algorithm.

**Strengths:**

1. The organization of the paper is easy to follow, although some technical parts are not clear (see Weakness 3).
2. The authors provide detailed simulation results to demonstrate the performance of the proposed algorithm. In particular, the authors present multiple variants of the proposed RCG algorithm with different graph inputs.

**Weaknesses:**

1. The main theoretical results in Sections 4 and 5 are based on the implicit assumption of atomic intervention (i.e., only one variable is affected by the failure mode). This is a very strong assumption, and existing methods such as RCD do not rely on this assumption. For example, in Figure 5, given that $F$ is not independent of $X_2$, it might be the case that both $X_2$ and $X_3$ are directly affected by $F$.

2. Section 5 lacks a theoretical guarantee of the recovery output, which may make the comparison with RCD unfair. It has been shown in RCD that the true root cause can be uniquely identified given infinite data (without knowing the graph structure or the number of root causes), although it may require an exponential number of invariance tests. The authors claim that only $O(n)$ invariance tests are needed in the RCG algorithm. However, there is no guarantee of recovery accuracy; that is, it is unclear under what conditions the true root cause is the only variable adjacent to $F$, as stated in Lemma 5.3.

3. Some technical details are either missing or provided in the appendix; including them in the main text would improve the presentation. For example, Lemma 5.3 relies on the possible parent set $PossPa(X)$, which is defined in Definition A.4 without explanation. Further, it appears that not all actual parents are possible parents (see Q1 below), which may lead to incorrect theoretical results.

**Questions:**

1. Consider a causal model with three variables $(X_1, X_2, X_3)$ with edges $X_1 \to X_2 \to X_3$ and $X_1 \to X_3$. Following the definition of possible parent in Definition A.4, $X_2$, which is an actual parent of $X_3$, is not a possible parent of $X_3$. Is this correct?

2. Given that the output of Algorithm 2 is a partially oriented DAG, is the definition of possible parent set the same as in Definition A.4?

3. Should the faithfulness assumptions be defined on the augmented graph?

---

> ### Author Response · Authors · 2024-11-21
> **Theoretical Guarantees**
>
> We thank the reviewer’s comments and questions. Below is our response.
>
> > The main theoretical results in Sections 4 and 5 are based on the implicit assumption of atomic intervention ...
>
> For the known graph case discussed in Section 4, we rely on the single root cause assumption as required by the baseline IGS algorithm. However, in the unknown graph case (Section 5), our approach does not depend on this assumption. For more details, please see our general comment.
>
> > Section 5 lacks a theoretical guarantee ...
>
> Our method, RCG, operates under the same set of assumptions as RCD in the unknown graph scenario. Lemma 5.3 establishes theoretical guarantees under these assumptions, showing that a root cause $X$ will have non-zero conditional mutual information with F, given its potential parents. In scenarios with multiple root causes, one can simply rank the conditional mutual information values in descending order to identify the top root causes. This ensures that our method can effectively handle multiple root causes in the unknown graph case.
>
> The key novelty of our algorithm lies in its ability to leverage a partial causal graph, learned during normal operation, to enhance root cause identification accuracy. This advantage is demonstrated in Figure 2, where our algorithm consistently outperforms the baseline RCD when provided with a valid C-essential graph.

---

> ### Author Response · Authors · 2024-11-21
>
> > Some technical details are either missing or provided in the appendix...
>
> Thank you for the suggestion. We will include more detailed explanations in future drafts. As per our definition, all parents are included in the set of possible parents. Please find our response to Q1 below.
>
> > Consider a causal model with three variables ...
>
> Thank you for the question. No, $X_2$​ _is_ included in the set of possible parents of $X_3$​. According to the definition provided in the manuscript, $X_2$ qualifies as a possible parent because there is no path from $X_3$ to $X_2$ that consists of an edge directed toward $X_2$.
>
> > Given that the output of Algorithm 2 is a partially oriented DAG, is the definition of possible parent set the same as in Definition A.4?
>
> Yes, it is. In fact, we introduce the concept of possible parents because we are working with an augmented graphical object rather than a traditional DAG.
>
> > Should the faithfulness assumptions be defined on the augmented graph?
>
> Since we assume the ground truth is a DAG, the faithfulness assumption ensures that all conditional independence (CI) relations can be derived from the DAG using d-separation. The augmented graph is introduced because, in practice, we can only recover a Markov equivalence class of the DAG based on the available CI constraints without further assumptions.
>
> We again, thank the reviewer for their thoughtful comments, questions, and interest in our work.

---

> > ### Comment · Reviewer_yWZE · 2024-11-25
> >
> > I thank the authors for the response. I would like to stay with my current evaluation of the work.
> > > Consider a causal model with three variables ...
> >
> > Here by "paths" do you mean only directed paths, as there is one path $X3 \gets X1 \to X2$? If only directed paths are considered, then I was wondering what is the difference between this set and non-descendant set.
> > > The augmented graph is introduced because we can only recover a Markov equivalence class of the DAG based on the available CI constraints without further assumptions.
> >
> > I think you were referring to CPDAG. My understanding is that augmented graph is introduced to represent interventions, which allows us to translate invariant distributions to independencies with F-node. In this case I believe a certain extension of conventional faithfulness assumption is required, as described in footnote 1 in the RCD paper.

---

> > > ### Author Response · Authors · 2024-11-25
> > >
> > > 1. Please note that according to Definition A.2, given a partially directed graph $D$, a \textit{path} from $V_{0}$ to $V_{n}$ in $D$ is a sequence of distinct vertices $\langle V_{0}, V_{1}, \ldots, V_{n} \rangle$ such that for $0 \le i \le n-1$, $V_{i}$ and $V_{i+1}$ are adjacent.
> > >
> > > 2. We apologize for the oversight in our previous answer. This is definitely correct! We do need a notion of an interventional faithfulness assumption in order to use F-node for further orienting the CPDAG. However, observe that we need a much weaker version than those previously used for learning interventional Markov equivalence classes. We will clarify this point in the camera ready, if the paper is accepted.
> > >
> > > We appreciate the reviewer's willingness to further engage with us and if they have any further questions we would be very happy to answer.

---

> > > > ### Comment · Reviewer_yWZE · 2024-11-26
> > > >
> > > > I am not sure if I may have misunderstood something, but according to Definition A.4,
> > > > > X is called a possible parent of Y if there is no path from Y to X that contains an edge with arrow pointing towards X.
> > > >
> > > > In this example, since there is one path $X_3 \gets X_1 \to X_2 $ that includes an edge with arrow pointing towards $X_2$, $X_2$ is not a possible parent of $X_3$. Please correct me if I am wrong.

---

> ### Author Response · Authors · 2024-11-26
>
> We apologize for mistakenly using paths instead of the edge between X and Y in the definition of possible parents. Here is an explicit list of edges we use to define Y being a possible parent of X in D.
>
> *PossPa(X):*
>
> - $Y\rightarrow X$
>
> - $Y o\rightarrow X$
>
> - $Y-X$
> - $Yo-o X$
>
> *NotPossPa(X):*
> - $X\rightarrow Y$,
> - $X\leftrightarrow Y$,
> - $X o\rightarrow Y$
> - X and Y are not adjacent.

---

### Author Response · Authors · 2024-11-21
**Single root cause assumption**

We sincerely thank the reviewers for their valuable feedback and efforts to improve the quality of our work. Here, we address one of the key concerns raised—the assumption in our method of a single root cause.

**Single root cause assumption:** Based on our discussions with site reliability engineers, we initially focused on cases where a single root cause was the primary consideration. However, after interacting with the reviewer's and carefully evaluating our claims, we are pleased to clarify that one of the central results of our paper (Lemma 5.3) naturally extends to scenarios with multiple root causes.

The single root cause assumption is required _only_ for Lemmas 4.2 and 4.3, which apply in the specific context where the complete DAG is known. However, as shown in our experiments, we observe that the sequential root-cause discovery approach (RCG(IGS)), which theoretically enjoys logarithmic number of CI tests, struggles in practice due to the finite number of samples. Consequently, Lemmas 4.2 and 4.3 primarily serve to establish the theoretical lower bound on the number of CI tests when the graph is fully known and a perfect CI oracle is available. In contrast, Lemma 5.3 offers a more practical solution that requires _only_ access to a partial causal graph and is sound even with multiple root causes.

---

### Author Response · Authors · 2024-11-27
**Manuscript Update**

We sincerely thank the reviewers for their valuable feedback, which has significantly improved the quality of our work. We have addressed the key concerns and incorporated most of the suggestions in the manuscript. Below is a summary of the changes we made:

- Added a brief discussion emphasizing cases with a single root cause and multiple root causes.
- Updated the definition of possible parent sets.
- Included the extended faithfulness assumption in the appendix.

Once again, we thank the reviewers and are happy to address any further questions.

---

### Meta-Review · Area_Chair_eYLB · 2024-12-21

**Metareview:**

This paper proposes an approach for root cause detection when the causal graph is unknown. All reviewers vote for rejection, and one notable expert pointed out some gaps in the claims made by the authors that need revision and clarification. The paper needs a major revision before it can be accepted.

**Additional Comments On Reviewer Discussion:**

There was an extensive discussion with reviewers, and one reviewer increased their score. The other reviewers maintained their score, mostly due to clarity concerns.

---

### Decision · Program_Chairs · 2025-01-22

Reject